

# Covariation of redox potential profiles and water table level in peatland sites representing different drainage regimes: implications for ecological modelling

Markku Koskinen[1], Jani Anttila[2], Valerie Vranová[3], Ladislav Holík[3], Kevin Roche[3], Michel Vorenhout[4,5], Mari Pihlatie[1], and Raija Laiho[2]

[1]University of Helsinki, Department of Agricultural Sciences, Viikinkaari 9, 00790 Helsinki, Finland
[2]Natural Resources Institute Finland, Latokartanonkaari 7, 00790 Helsinki, Finland
[3]Mendel University in Brno, Department of Geology and Soil Science, Faculty of Forestry and Wood Technology, Zemedelska 3, 613 00 Brno, Czech Republic
[4]Institute for Biodiversity and Ecosystem Dynamics—Freshwater and Marine Ecology (IBED-FAME), University of Amsterdam, P.O. Box 94240, 1090 GE Amsterdam, The Netherlands
[5]MVH Consulting, 2317 BD Leiden, The Netherlands

**Correspondence:** Markku Koskinen (markku.koskinen@helsinki.fi)

**Abstract.** Reduction-oxidation (redox) reactions are ubiquitous in nature, responsible for the energy acquisition of all organisms. Redox reactions are electron transfer reactions and necessarily involve two participants: one being oxidised (electron donor) and one being reduced (electron acceptor).

Availability of terminal electron acceptors (TEAs) is a major determinant of the extent to which the carbon in OM can be
oxidised in an ecosystem. This is the most important under waterlogged conditions, such as in peatlands, where diffusion of $O_2$, the most effective common TEA, into the soil is blocked by water. Under these conditions, available alternate TEAs are used by microbiota to continue OM oxidation.

The decomposition processes in soil can be characterised by its redox state, ie. which TEA is responsible for oxidation of OM at a given time. This can in principle be measured as a voltage between the soil solution and a known reference electrode,
known as the redox potential.

Current soil ecosystem models do not depict the use of alternate TEAs well. This limits their applicability for predicting soil carbon loss under different drainage regimes, and thus their usefulness for assessing best management practices for soil carbon preservation and water course protection. Water table level (WTL) is the most common determinant of the mode of decomposition in ecosystem models, implying the assumption that the redox state of a peatland ecosystem responds predictably
to changes in WTL.

We conducted a two-year redox monitoring experiment in a boreal mesotrophic peatland under three drainage regimes: undrained, short-term drainage and long-term drainage. In addition, an ombrotrophic long-term drained plot was monitored. Snapshot assessments of the activities of three major metabolic enzymes, arginine deaminase, protease and urease, were also done in the mesotrophic plots, indicating differences in microbial activities between the drainage regimes.




We found that WTL was a poor temporal predictor of redox potential, but that the position of major transition zones between oxic and anoxic states as well as enzymatic activities within the peat profile were somewhat determined be the dominant WTL depth. In the undrained plots especially redox potential values reflecting oxic or suboxic conditions were often found below the WTL, whereas on the drained plots anoxia was present above the WTL. Preceding redox potential was found to affect enzymatic activities of protease and urease, but not arginine, in all measured plots.

## 1   Introduction

Oxidation-reduction (redox) reactions are central to the energy acquisition processes of all life. All redox reactions consist of an atom donating an electron (being oxidised) and an atom accepting an electron (being reduced). In soils, there is usually no shortage of electron donors, the most usual source being organic matter; thus, energetically feasible redox reactions are generally limited by which electron acceptors are available (Green and Paget, 2004).

Many elements found in soils alter their behaviour according to their redox state. As an important example, iron (Fe) is mainly found in non-water soluble compounds in its oxidised ferric $Fe^{3+}$ (Fe(III)) state, whereas in its reduced ferrous $Fe^{2+}$ (Fe(II)) state it is soluble. This has implications for the movement of elements in soil solution. For example, phosphorus (P) forms complexes with Fe(III) compounds that may then 'unravel' when Fe is reduced, making the P available to soil solution (e.g. Zak et al., 2004).

The strongest common oxidising element in soil systems is oxygen ($O_2$). This is consumed whenever available, and such reactions release the most energy. When availability of $O_2$ is limited, e.g. under waterlogged conditions, other terminal electron acceptors (TEAs) will be used. Common TEAs in soils include, in descending order of available energy, nitrate ($NO_3^-$), manganese(III) ($Mn^{3+}$), Fe(III), sulphate ($SO_4^{2-}$) and carbon dioxide ($CO_2$), each of which require increasing electron activity (pe) in the soil solution to be energetically feasible. This can be measured against a redox pair of known activity to provide the redox potential ($E_h$, V).

Redox processes in soils, whether mineral or organic, are an interplay between microbes, their enzymes and purely chemical reactions, i.e. both biotic and abiotic drivers. Which process is dominant will depend on the relative availability of oxidants and the activity of $H^+$, i.e. the environment's pH. For example, Fe(III) may be reduced to Fe(II) either biotically, by microbes oxidising organic carbon (C), or chemically, by reduced (i.e. electron-rich) humic substances (Melton et al., 2014). In more extreme circumstances, Fe(III) may also be reduced by archaea in the anoxic oxidation of $CH_4$ (Ettwig et al., 2016). As microbe activity is often the dominant factor affecting redox status, it is conceivable that changes in $E_h$ could be more rapid under conditions where microbes are more active, reflected by microbial enzyme activity in the soil.

In peatlands, the activity of oxidative and hydrolytic enzymes produced by microorganisms will be regulated by the site's vegetation cover, soil water regime, temperature and nutrient availability, along with interactions between physicochemical factors, such as changing pH or redox $E_h$ (Freeman et al., 1996; Bonnett et al., 2006). Interactions between redox conditions, extracellular enzyme activity and persistence of phenolic substances are complex. For example, Freeman et al. (2001) hypothesised that a process termed "enzymatic capture" occurs due to the accumulation of phenolic substances (phenolic dis-



solved organic matter, or phenolic DOM) and their persistence in the soil. This condition occurs when phenoloxidase activity
is suppressed under anoxic conditions. The accumulation of these phenolic substances leads to inhibition of hydrolytic enzyme
activity, e.g. of protease or urease (Kane et al., 2019), and has been suggested as one mechanism leading to stabilisation of peat
C (Freeman et al., 2004).

Humic substances, prevalent in organic soils such as peat, have been found to act as both TEAs and donors, potentially re-
ducing $CH_4$ emissions from boreal peatlands by a large factor (Klüpfel et al., 2014). The $E_h$ of humic acid reduction reactions,
for example, has been shown to range between +150 – -300 mV, thus overlapping with several non-organic electron acceptors
(Aeschbacher et al., 2011). Furthermore, the electron accepting capacity of organic matter has been found to be greater in
ombrotrophic than minerotrophic peats (Keller and Takagi, 2013). Finally, other non-$O_2$ electron acceptors have been found to
inhibit production and/or promote oxidation of $CH_4$ (Kumaraswamy et al., 2001).

Generally speaking, current ecological models used for predicting ecosystem C and nutrient fluxes either only include an
oxic-anoxic state controlled by soil permeability and water table level (WTL) (Laurén et al., 2016), ignore redox altogether
and base their estimates on WTL and/or soil moisture and temperature only (e.g. Gong et al., 2013) or use only one electron
acceptor other than $O_2$, e.g. $NO_3^-$ (Wriedt and Rode, 2006) or Fe(III) (Tang et al., 2016). To improve such models, reactions
of TEAs other than $O_2$ need to be modelled. A first step in this direction would be to model the redox state of a peatland
empirically, using easily measurable (and computationally cheap) variables and site properties to predict the momentary $E_h$.

Northern peatland ecosystems generally have high WTLs, which has encouraged the sequestration of huge amounts of C
under anoxic conditions (e.g. Yu et al., 2010). At such sites, any variation in soil redox $E_h$ due to changes in the WTL could
unlock the stored C into water-dissoluble and microbially degradable forms (Freeman et al., 2001). Indeed, the ecological char-
acteristics of such wet and relatively nutrient-rich fens, as well as the C stored in the peat, have been shown to be particularly
sensitive to any decrease in WTL (Straková et al., 2012; Jaatinen et al., 2007; Gong et al., 2013; Kokkonen et al., 2019).

It is generally believed that redox conditions in peatlands are regulated by the WTL through its effect on soil $O_2$ concentra-
tions, i.e. that in water-logged soil, the water blocks gas movement between the atmosphere and the soil matrix (e.g. Belyea,
1999; Blodau et al., 2004; Kiuru et al., 2022). However, covariation of WTL and soil redox $E_h$ at different depths in peat soils
has not been examined in detail. In dense peat especially, the depth distribution of redox processes may be insensitive to WTL
fluctuations (Knorr and Blodau, 2009). On freshwater tidal wetlands, for example, redox $E_h$ has been shown to fluctuate at 20
cm depth, but not at 50 cm, despite the WTL fluctuating between -40 and +18 cm (Seybold et al., 2002).
In recent years, there has been growing interest in defining the best approach for describing redox phenomena in ecosystems.
For example, it has been suggested that only assessing the $E_h$ state of an ecosystem or soil profile should be amended by
examining the resistance of that system to change under redox conditions, i.e. its redox buffering capacity (Burgin and Loecke,
2023). In such cases, it could be postulated that:

1. TEAs other than $O_2$ and $CO_2$ may play a significant role in an ecosystem's redox palette; and

2. the readings given by $E_h$ measurement devices reflect the dominant redox pair in an ecosystem at a given time; then

3. under increasingly anoxic conditions, $E_h$ should "pause" its descent at values reflecting the available TEAs; and



4. this would cause the probability distribution of $E_h$ values measured in such an ecosystem to express multi-modality rather than bi-modality, which would be indicative of an $O_2$-$CO_2$-dominated system.

From a modelling perspective, this would indicate that current peatland models that do not recognise TEAs other than $O_2$
would work better on ombrotrophic than minerotrophic sites.

In this study, we examine the effects of WTL fluctuation on soil solution redox $E_h$ at different depths in a mesotrophic (ME) sedge fen site situated in a minerotrophic peatland in southern Finland. Four plots were examined, comprising three ME plots with different soil WTL (drainage) regimes, i.e. wet (undrained control), short-term water-level drawdown (15 years), and long-term drawdown (55 years), along with a long-term drained bog plot at an ombrotrophic (OM) site in the same peatland
massif, used for comparison. In each case, WTL drawdown has had marked impacts on local peat properties (e.g. chemical composition, density, pH), vegetation cover and soil microbial communities.

We hypothesise that the drainage regime, and the ensuing depth range of WTL fluctuation, will be reflected in the depth at which redox conditions are at their most dynamic, i.e. at ME plots, with the wet undrained plot having WTL fluctuations and redox dynamics occurring close to the peat surface, the long-term drained plot having fluctuations at the greatest depth, and
the short-term drained plot lying somewhere between.

We also tested for relationships between WTL and $E_h$ under the different drainage regimes using analysis of wavelet coherence between WTL and $E_h$ conditions, using both $E_h$ values from individual sensors at different depths and the depth isopotential of 0mV $E_h$, indicative of Fe reduction. We hypothesise that $E_h$ from sensors in the most active peat layer in terms of alternating redox conditions will follow fluctuations in the WTL, and that the 0 mV $E_h$ isopotential will consistently follow
the WTL.

Secondly, we hypothesise that the lower levels of alternate TEAs in the OM plot will cause $E_h$ conditions to become reducing more quickly after a rise in WTL than in the ME plots. This will be revealed as a lower phase difference, i.e. lag time, in the wavelet coherence analysis in the OM plot compared to the ME plots, and as a more extreme distribution of $\delta E_h$ in the long-term drained plot compared to the wet, undrained plot.

Thirdly, we hypothesise that the lack of alternative TEAs at the OM plot will be expressed as bi-modality in the probability density distribution of $E_h$ measurements, and that the presence of alternative TEAs at the ME plots will be expressed as multi-modality.

Finally, we test whether 'snapshots' of soil enzyme activity status at the ME plots correlate with preceding redox conditions and drainage status. We hypothesise that the more aerobic conditions on drained plots will be reflected in higher enzyme
activity compared to that at the wet undrained plot.



## 2 Material and methods

### 2.1 Measurement sites

This study took place at the Lakkasuo mire, a raised bog complex with a large minerotrophic lag, in southern Finland (61.797 °N, 24.309 °E). In 1961, half of the mire complex was drained to improve forest growth, which created conditions where
the impact of persistent WTL drawdown could be studied on plots either side of a border ditch, both of which shared similar hydrological conditions, vegetation cover and soil properties prior to drainage (e.g. Minkkinen et al., 1999). The surface layer of the undrained minerotrophic fen receives ground water from the Vatiharju esker bordering the mire complex in the west, potentially bringing in electron acceptors such as Fe to the mesotrophic (ME) plots.

In 2001, several new experimental WTL drawdown plots were established, with conditions allowing the study of shorter-
term impacts (e.g. Straková et al., 2012; Kokkonen et al., 2019). These measurement plots were positioned to represent three WTL regimes, i.e. an undrained plot (ME-UD), a short-term (ca. 15 yrs at the time of study) persistent WTL drawdown plot (ME-STD) and a long-term (ca. 55 yrs) persistent WTL drawdown plot (ME-LTD) (Fig. 1). Conditions at the STD plot were maintained via narrow, ca. 30-cm deep ditches that conducted surface water into the border ditch. An initially ca. 1m deep ditch network within the LTD area had not been intensively reconditioned for some years, thus the drainage effect was less effective
than would be the case at more intensively managed forest sites. In addition, a comparative ombrotrophic (OM) plot under an LTD regime (OM-LTD) was located on the same peatland massif, ca. 500 metres from the ME LTD plot.

The different WTL regimes being examined were reflected in the composition of plant communities on the plots. For example, The ME-UD plot supported typical boreal mesotrophic open mire vegetation, being dominated by sedges, such as *Carex lasiocarpa* (woollyfruit sedge) and *C. rostrata* (beaded sedge), with some forbs, such as *Menyanthes trifoliata* (bog
bean) and *Comarum palustre* (swamp cinquefoil), shrubs *Betula nana* (dwarf birch) and *Salix* sp. (willows), and several moss species, the most common being peat mosses *Sphagnum fallax* and *S. flexuosum*. On the ME-STD plot, the lowered WTL had already had a significant effect on the vegetation (Kokkonen et al., 2019), the original sedges and forbs largely having been replaced by species tolerating drier conditions, such as *C. echinata* (star sedge), *C. canescens* (silvery sedge), *Calamagrostis* spp. (reed grass), *Cirsium palustre* (European swamp thistle) and *Trientalis europaea* (arctic starflower). A dense spread of
*Pinus sylvestris* (Scots pine) and *Betula pubescens* (Downy birch) seedlings had started to form an overstorey layer and, while the moss layer was sparse, it included several species. The ME-LTD plot could be classified as a mesotrophic peatland forest (*Vaccinium myrtillus* type II in the local classification), with surface vegetation characterised by shrubs, such as *Empetrum nigrum* (black crowberry), *V. myrtillus* (bilberry) and *V. vitis-idaea* (lingonberry), with *Eriophorum vaginatum* (cottongrass) as the most common sedge. The tree stand comprised *P. sylvestris* and *B. pubescens*, with some *Picea abies* (Norway spruce),
while the moss layer was dominated by *Pleurozium schreberi* (feathermoss), other moss species including the peat mosses *S. russowii*, *S. medium* and some *Dicranum* (fork mosses) and *Polytrichum* (haircap moss) species. In comparison, the OM-LTD plot was classified as a nutrient-poor peatland forest, with *Sphagnum* mosses intermixed with forest mosses, and peatland shrubs such as *Andromeda polifolia* (bog rosemary) sparsely present among *E. vaginatum* tillers.



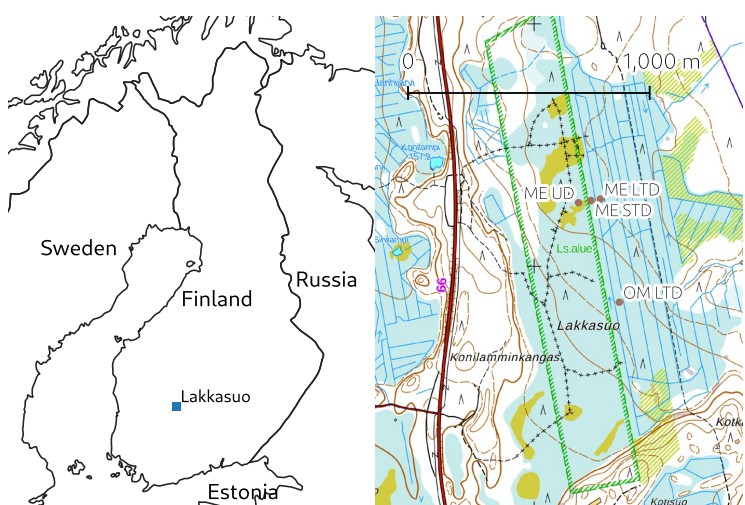

**Figure 1.** Left: location of the measurement site (made with Natural Earth); right: map of the Lakkasuo mire (north at top), with the measurement plots marked (background map: National land survey of Finland, topographic map, 4/2024. Green hatched rectangle is a feature of the topographic map defining the perimeter of a nature conservation area.)

Soil chemical properties (Laiho et al., 2024) at the ME plots reflected the typical effects of drainage on fen peats, *i.e.* a relative
increase in C content and a decrease in nitrogen (N) content, along with a decrease in pH due to decomposition increasing the amount of organic acids in the soil, and loss of soluble earth metals such as magnesium (Mg) and potassium (K) (Fig. 2). From a redox perspective, the most important elements analysed were Fe and manganese (Mn), with Fe content being significantly lower in the surface layer of LTD plots than STD and UD plots, Mn content showing no significant differences. While a soil chemical analysis was not available for the OM-LTD plot, concentrations of the most important elements for redox couples
(i.e. Fe and Mn) on drained OM-oligotrophic peatlands are known to be around one-fifth those present on drained mesotrophic sites (e.g. Laiho and Laine, 1995).

## 2.2 Redox sensors

Three redox probes (Vorenhout consulting, Netherlands) fitted with platinum (Pt) electrodes were installed at five depths (i.e. 5, 15, 25, 35 and 45 cm below the surface) at each measurement plot in early Autumn 2014. A silver-silver chloride
(Ag-AgCl) reference electrode was also installed in a groundwater well for each trio of probes. A Hypnos III logger system (Vorenhout Consulting, Netherlands) (Vorenhout et al., 2011) was then used to record data from the probes at 15-minute intervals continuously over two years, i.e. from Autumn 2014 to Autumn 2016.



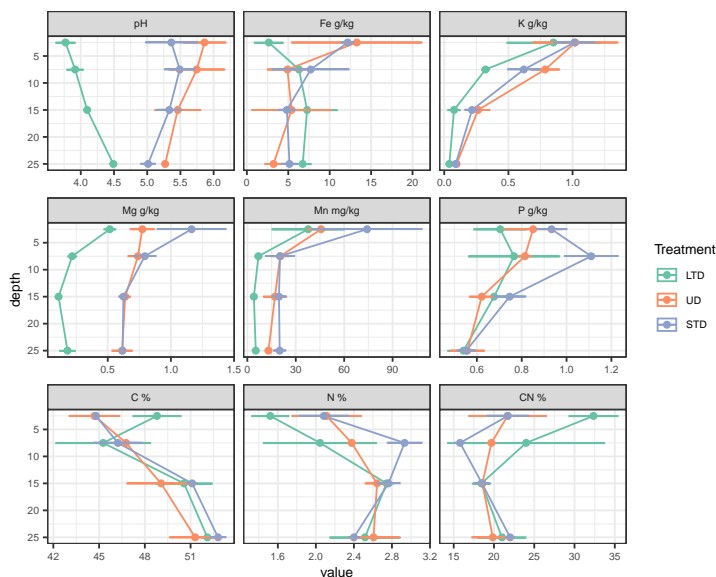

**Figure 2.** Chemical properties of peat soils at the three mesotrophic study sites (means and standard deviation of three samples; element concentrations in mg or g/kg dry weight unless stated otherwise in the panel headers; depth in cm. LTD (green) = long-term drawdown, UD (red) = undrained, STD (blue) = short-term drawdown).

**Table 1.** Common redox pairs in soils and their corresponding field-observable $E_h$ values (mV).

| | Redox pair | |
|---|---|---|
| $E_h$ range | Oxidised species | Reduced species |
| > 400 | $O_2$ | $H_2O$ |
| 300 | $NO_3^-$ | $N_2$ |
| 150 | $Mn^{4+}$ | $Mn^{3+}$ |
| -50 … 100 | $Fe^{3+}$ | $Fe^{2+}$ |
| -150 | $SO_4^{2-}$ | $H_2S$ |
| <-200 | $CO_2$ | $CH_4$ |





### 2.3 Ancillary measurements

The WTL at each measurement plot was continuously monitored using TruTrack WT-HR1000 probes (Intech Instruments, New Zealand), while soil temperatures were monitored using i-Button DS1921G temperature loggers (MaximIntegrated Products, USA) at the same depths as $E_h$ (i.e. 5, 15, 25, 35 and 45 cm) at one point per treatment.

Soil pH was determined for each depth at each ME and OM plot on two occasions, October 2014 and May 2015, from peat samples obtained from the plots. The pH was measured under laboratory conditions using a slurry of one part peat to three parts water.

Activity of the enzymes arginine deaminase, protease and urease were determined at ME plots as indicators of soil microbial activity at the same depths as redox measurements on peat samples obtained in October 2014, May 2015 and July 2015. Briefly, arginine deaminase activity was determined from soil samples incubated with arginine to produce ammonium ($NH_4$). This was then extracted with potassium chloride (KCl) solution and the amount of extracted $NH_4$ determined colourimetrically (Alef and Kleiner, 1986). Determination of protease activity was based on the decomposition of added casein during incubation, determined colourimetrically by measuring the amount of L-tyrosine produced (Rejsek et al., 2008). Urease activity was determined by incubating the samples with added urea and colourimetrically determining the amount of $NH_4$ released (Kandeler and Gerber, 1988).

Air temperature was measured continuously at 2m above the soil's surface near the measurement plots throughout the measurement campaign. Precipitation and snow depth were measured hourly at the SMEAR II measurement station, ca. 6 km distant (Aalto et al., 2023).

### 2.4 Data processing

The potential given by the Pt-Ag/AgCl-pairs was converted to $E_h$ by adding +200mV to the reading. The readings were then corrected for $H^+$ activity by applying the Nernst equation, i.e. $(+59mV \times (pH - 7))$. To reduce noise, the readings were averaged over an hour or over a day, depending on the analysis being undertaken. The data were further corrected for drift by fitting a linear model of $E_h$ over time to the maximum (> 400 mV) and minimum (< 400 mV) readings, as grouped by the logger. Drift was determined at $-0.16mVd^{-1}$ for the STD and LTD plots and $-0.08mVd^{-1}$ for the UD and OM-LTD sites.

Basic data processing and statistical analyses were undertaken using the R statistical package (R Core Team, 2023).

### 2.5 Linear and non-linear modelling

Linear and non-linear models, using hourly data, were fitted to assess differences in $E_h$ response to WTL between the ME measurement plots. Both the models and the results are described in more detail in the supplementary materials.

### 2.6 Wavelet analysis

The frequency spectra of hourly averaged redox potentials, soil temperatures and WTL measurements were analysed using wavelet decomposition (Torrence and Compo, 1998) to identify patterns that would inform future directions for $E_h$ modelling.



Briefly, wavelet decomposition consists of fitting a wavelet function with different frequencies at each time point in a time
series and estimating how well each of the frequencies fits the data, enabling identification of time patterns with above- and
below-average values.

The coefficients were then compared against soil temperature and WTL changes *via* a cross-correlation spectrum, using a
*Mexican hat* style mother wavelet to achieve sufficient temporal resolution (Wang, 2015). Owing to large differences between
the probe $E_h$ readings, wavelet fitting was undertaken separately for each probe and depth profile. All hourly wavelet analyses
were performed using the PyWavelets package in Python (Lee et al., 2019). The wavelet coherence figures are presented in
Supplement 1.

In addition, wavelet coherence between daily average WTL and the depth isopotential of $E_h$ 0 mV indicative of Fe reduction
was estimated based on the mean $E_h$ of all three probes on all measurement plots using the WaveletComp package in R
(Schmidbauer, 2018). In this case, the depth isopotential was calculated by linearly integrating daily average $E_h$ depth profiles
from 5 to 45 cm depth, using a random time series with the same Fourier transform properties as a surrogate time series to
analyse the significance of coherence. Only those days where the $E_h$ 0 isopotential was present (i.e. the whole $E_h$ profile was
not above or below 0 mV) were included in the analysis.

## 3   Results

### 3.1   Water table level

Differences in WTL were significant ($p < 0.05$) between all treatments, with mean WTLs over the measurement campaign
being -1.5 cm, -13.8 cm and -26.8 cm for the ME-UD, ME-STD and ME-LTD plots, respectively (Fig. 3). Fluctuations in
WTL were greatest at the LTD plot, while the STD and UD plots showed roughly the same magnitude of WTL fluctuation, i.e.
between 0 and -15 cm.

Wavelet analysis indicated clear seasonal patterns in WTL fluctuation for the UD and LTD plots, but not for the STD plot.
This pattern was especially strong in the LTD plot, which showed below average values from July to November and above
average values from January to May (Fig. 4). At the UD plot, WTL fluctuated over a shorter period than at the LTD plot (UD
1000-hour period vs. LTD 2000-hour period) (Figs. 5, 6). In comparison, temperature plots at all sites showed the opposite
pattern.

### 3.2   Redox

Generally speaking, $E_h$ varied between oxic and completely anoxic in all profiles, with the 5 cm surface layer being over $+400$
mV and layers below 25 cm being $< 0$ mV for all three treatments. The main difference between treatments was at 15 cm depth,
where $E_h$ was almost permanently below $-200$ mV after June 2015 at the UD plot, but varied between $+250$ mV and $-400$
mV in the STD and LTD plots (Fig. 3).

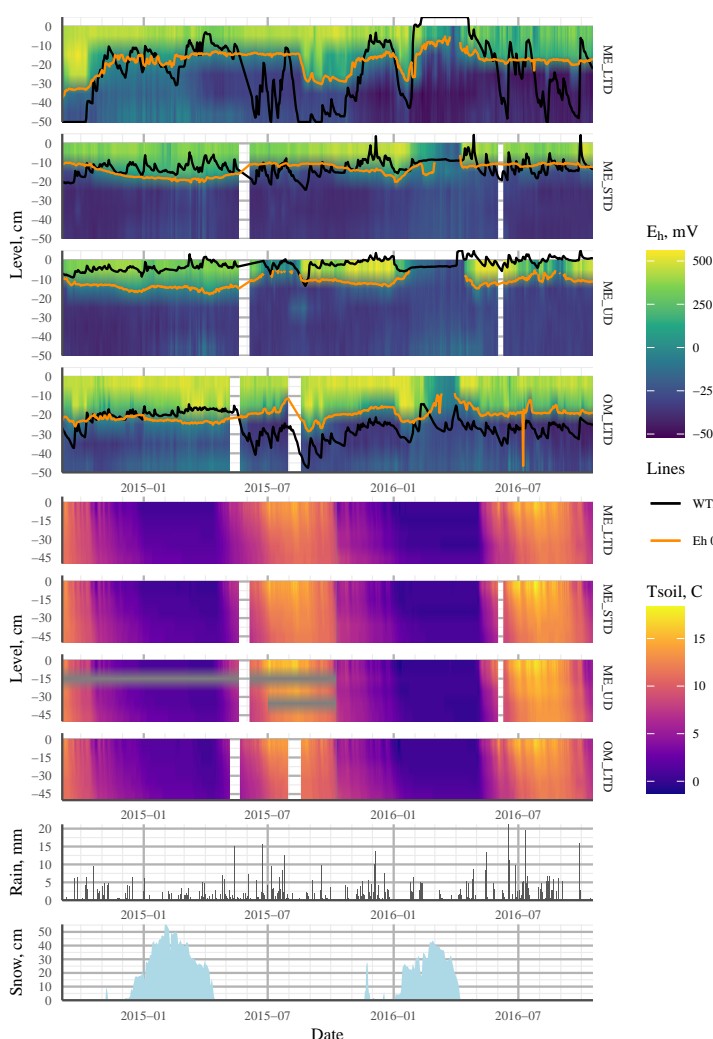

**Figure 3.** Heat maps showing mean values (mean of three profiles per treatment) interpolated from the 5-45 cm depth profiles of $E_h$ (four topmost plots; ME LTD, ME STD, ME UD, OM LTD) and soil temperature (four middle plots; ME LTD, ME STD, ME UD, OM LTD), with WTL and the $E_h$ 0 isopotential shown as solid black and orange lines over the $E_h$ plots, respectively. The two plots at the bottom show daily rainfall (mm) and snow pack thickness (cm) at the peatland over the study period. ME = mesotrophic sedge fen; OM = ombrotrophic bog; UD = undrained; STD = short-term ($\sim$ 15 years) drainage; LTD = long-term ($\sim$ 55 years) drainage.



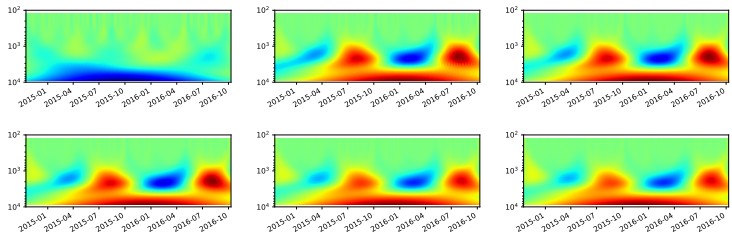

**Figure 4.** Top to bottom, left to right: Mexican hat wavelet coherence with 1) WTL data from the ME-STD plot; 2) soil temperature (soilT) at 5 cm depth; 3) soilT at 15 cm depth, 4) soilT at 25 cm depth; 5) soilT at 35 cm depth; 6) soilT at 45 cm depth.

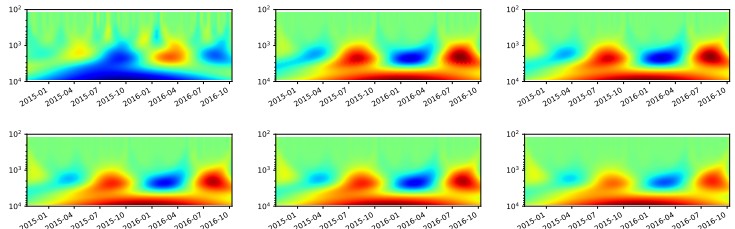

**Figure 5.** Top to bottom, left to right: Mexican hat wavelet coherence with 1) WTL data from the ME-LTD plot; 2) soil temperature (soilT) at 5 cm depth; 3) soilT at 15 cm depth, 4) soilT at 25 cm depth; 5) soilT at 35 cm depth; 6) soilT at 45 cm depth.

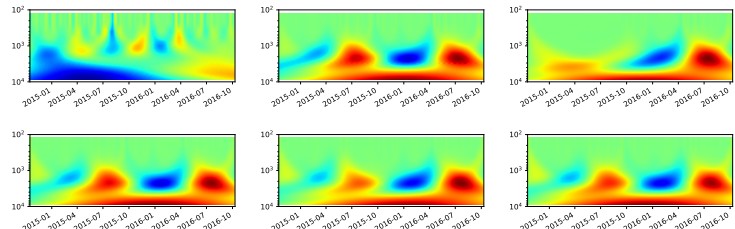

**Figure 6.** Top to bottom, left to right: Mexican hat wavelet coherence with 1) WTL data from the ME-UD plot; 2) soil temperature (soilT) at 5 cm depth; 3) soilT at 15 cm depth, 4) soilT at 25 cm depth; 5) soilT at 35 cm depth; 6) soilT at 45 cm depth.



While $E_h$ values varied greatly between the three profiles for all treatments, especially at 15cm depth (Fig. 3), sensitivity to change in WTL differed between profiles, with some sensors showing no sensitivity to WTL and others consistently reacting to WTL change.

Wavelet analysis revealed a periodic pattern of roughly 100 days (2000-3000 hours) at differing depths on all plots. At the ME drained plots (STD, LTD), for example, clearest patterns were observed at 15 and 25 cm depth (Figs. 7, 8, 9), whereas strongest patterns were observed at 5 cm depth at the UD plot (Fig. 10). There were also additional shorter-period patterns at the UD plot ranging from 200 to 500 hours. At the STD plot, interactions between $E_h$ and WTL in the most active layers were mostly in synphase, i.e. a lowering WTL (increasing WTL depth) coinciding with rising $E_h$ values, compared with an alternating counter- and synphase interaction at the UD and LTD plots.

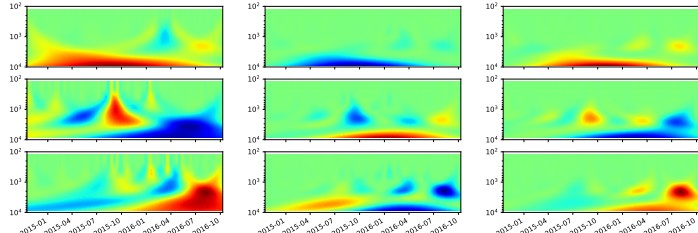

**Figure 7.** Left: Wavelet coherence between Mexican hat wave and $E_h$ at ME-LTD plot, 15 cm depth; center: interaction between $E_h$ and WTL wavelets; right: interaction between $E_h$ and soil temperature wavelets at LTD plot, 15 cm depth. 3 probes plotted separately.

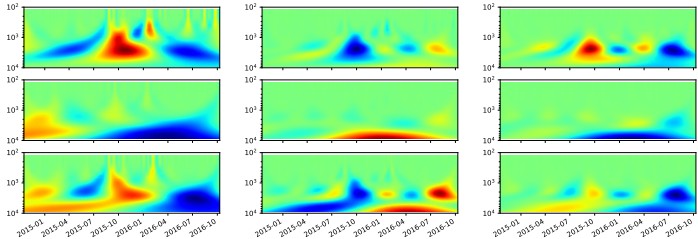

**Figure 8.** Left: Wavelet coherence between Mexican hat wave and $E_h$ at LTD plot, 25 cm depth; center: interaction between $E_h$ and WTL wavelets; right: interaction between $E_h$ and soil temperature wavelets at ME-LTD plot, 25 cm depth. 3 probes plotted separately.

Analysis of coherence between the Fe-reducing depth isopotential (-100 ... +50 mV (Table 1), represented by the $E_h$ 0 mV isopotential) and WTL (Fig. 11) indicated 128-day and 64-day periods of significant (p < 0.05) coherence at all plots, but particularly so at the LTD plot, where WTL led the change during these periods. At the ME-STD and ME-UD plots, there were fewer periods of significant coherence and phase differences alternated between WTL and the $E_h$ 0 isopotential leading.





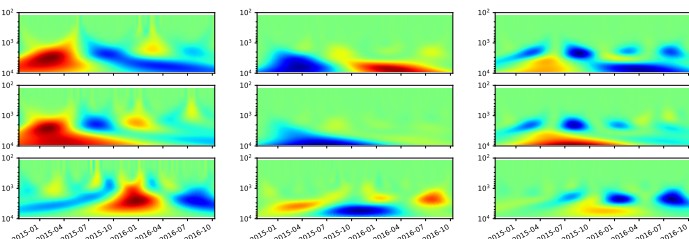

**Figure 9.** Left: Wavelet coherence between Mexican hat wave and $E_h$ at STD plot, 15 cm depth; center: interaction between $E_h$ and WTL wavelets; right: interaction between $E_h$ and soil temperature wavelets at ME-STD plot, 15 cm depth. 3 probes plotted separately.

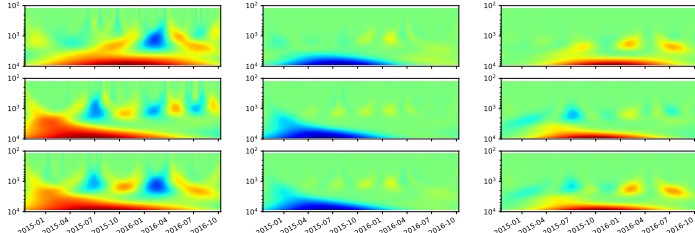

**Figure 10.** Left: Wavelet coherence between Mexican hat wave and $E_h$ at the UD plot (5 cm depth); centre: interaction between $E_h$ and WTL wavelets; right: interaction between $E_h$ and soil temperature wavelets at the ME-UD plot (5 cm depth). Three probes plotted separately.

During the first winter, $E_h$ remained above Fe-reducing levels in soil > 15 cm depth at all plots, regardless of WTL fluctuation and snow pack height. During the second winter, however, $E_h$ values indicative of $CO_2$ reduction were achieved at all plots, even at 5 cm depth (Fig. 3).

A comparison of $E_h$ probability densities at 25 cm depth on the ME-LTD and OM-LTD plots (where most $E_h$ dynamics occurred) indicated bi-modality between $O_2$ reduction and $CO_2$ reduction at the OM-LTD plot and multi-modality, with peaks at $E_h$ levels corresponding to $O_2$, $Fe^{3+}$ and $CO_2$ reduction, at the ME-LTD plot (Fig. 12). Inundation of the sensor had a more pronounced effect at the ME plot, shifting the distribution toward lower $E_h$ values.

### 3.3 Enzyme activities

Monitoring of enzyme activity at the ME site in Autumn 2014 revealed a significant decrease in protease and urease activity with depth at all plots (LTD, STD, UD); however, the rate at which activities decreased differed between plots (Table 2). At the control plot (UD), both protease and urease were found in high levels in the topsoil, decreasing rapidly down to 20–30 cm, after which the rate of decline slowed. At the STD plot, subsoil (40-50 cm depth) protease levels were ca. 4x, and urease ca. 6x, lower than those in the topsoil (0-10 cm depth), and the rate of decline in both cases was less steep than that at UD plot,

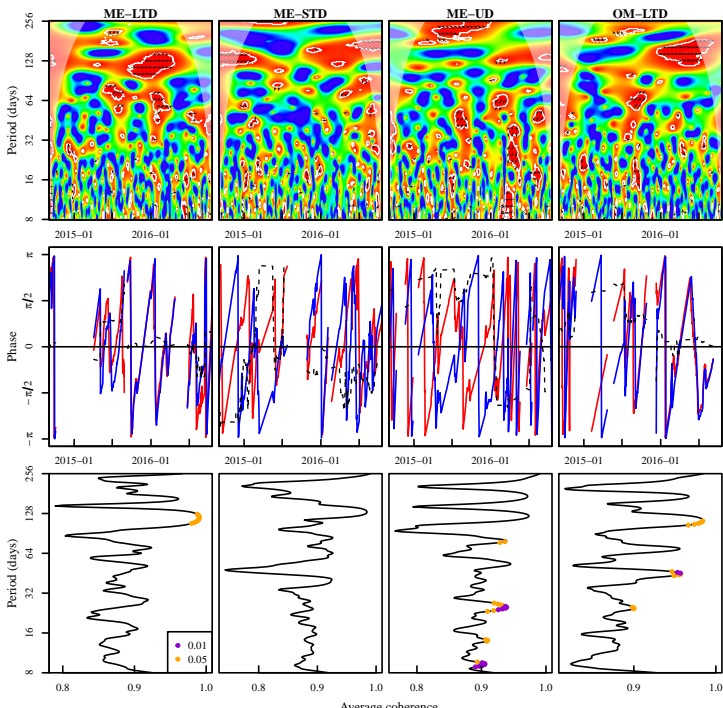

**Figure 11. Top row:** Wavelet coherence between the $E_h$ 0 depth isopotential and WTL (blue areas = low coherence, red areas = high coherence, areas of significant coherence ($p < 0.05$ vs random time series with identical fourier transform characteristics) outlined in white; black arrows = phase difference: pointing right = synphase interaction, pointing left = counterphase interaction; right-up or left-down = WTL leads, left-up or right-down = $E_h$ 0 depth isopotential leads; areas shaded with white = cone of influence). **Middle row:** phases and phase differences during periods of significant coherence (blue line = phase of $E_h$ 0 depth isopotential, red line = WTL phase, dashed black line = phase difference: $-\pi \ldots -\pi/2$ and $0 \ldots \pi/2$ = WTL leads, $-\pi/2 \ldots 0$ and $\pi/2 \ldots \pi$ = $E_h$ 0 isopotential leads). **Bottom row:** average coherence (x-axis) per period length (y-axis), with period length of significant coherence denoted with orange and dark violet points ($p < 0.05$ and $p < 0.01$, respectively). Measurement plots indicated in top row panel titles; x-axes aligned between panels in first and second rows.



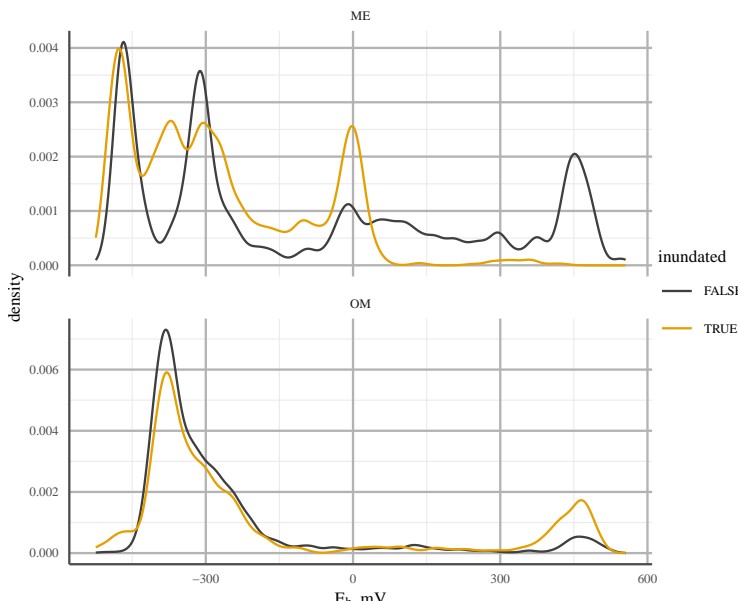

**Figure 12.** Probability density of $E_h$ values at 25 cm depth for mesotrophic (ME; top) and ombrotrophic (OM; bottom) long-term drainage (LTD) plots. Line colour indicates whether WTL was above (orange) or below (black) the sensor. Note: different y-axis scales used due to a more continuous distribution at the ME site.

tending to level off at around 30–40 cm. While 40-50 cm depth urease levels were similar to those at UD plots, protease levels at the same depth were significantly lower. At LTD plots, while urease showed a gradual but significant decline with depth, with levels around half those at UD plots at 40–50 cm, protease levels first declined but then began increasing again at around 20 cm, reaching similar levels to topsoil at 30–40 cm. After remaining relatively stable down to 10–20 cm, levels of arginine deaminase showed increasing trends with depth at all plots, with significantly higher levels in deeper soils at UD (30–40 cm)

and STD (40–50 cm) plots, but not at LTD plot (Table 2). Arginine levels at the STD plot were higher than those at the UD and LTD plots at all depths, with especially high levels at 40–50 cm. Overall, the rate of increase was steepest at the UD plot, with levels at 40–50 cm 2.5x higher than those in topsoil, followed closely by the STD plot, with a significant increase at 40–50 cm ca. 2.1x higher than that in the topsoil. In comparison, the LTD plot showed a much shallower rate of increase, with levels at 40–50 cm just ca. 0.5x higher than those in topsoil.

In Spring 2015, both the levels and activity patterns for protease were similar to those in Autumn 2014, with a relatively steep and significant decline with depth at the UD plot (though the rate of decline was slower than 2014 below 20–30 cm), a steep decline at the STD plot that increased slightly at 40–50 cm, and a steep and significant decline at the LTD plot down to



20–30 cm, after which levels increased steeply, approaching topsoil levels again at 40–50 cm (Table 3). In comparison, topsoil urease levels were noticeably higher than in 2014 at all three plots, and especially so at the STD (ca. 2x) and LTD (ca. 3x) plots. However, levels dropped rapidly with depth at all plots, levelling off at 30–40 cm at plot UD at higher levels than 2014; dropping but peaking significantly at 40–50 cm at the STD plot at levels slightly higher than 2014; and dropping but peaking significantly at 30–40 cm at the LTD plot, with final levels slightly lower than in 2014. The greatest change between Spring 2015 and Autumn 2014 was observed in arginine deaminase levels, with levels significantly lower in 2015 at all plots (UD topsoil ca. 4.5 x lower, STD ca.3.5x lower, LTD ca. 4x lower). All three plots showed a further decline to 10–20 cm (LTD = 20–30 cm) followed by a significant increase up to 40–50 cm (LTD = 30–40 cm followed by a further drop). Levels at 40–50 cm were between 7 and 8x lower than in 2014.

Overall, enzyme activity in Summer 2015 (Table 4) was similar to that in Spring (Table 3), with the difference that topsoil protease levels were slightly higher at UD and STD plots, and slightly lower at the LTD plot; urease topsoil levels were slightly lower at UD and STD, and slightly higher at LTD; and topsoil arginine levels were relatively similar at UD and LTD but slightly lower at STD (Table 4). In all cases, levels at depth (40–50 cm) were slightly lower than Spring at UD and LTD plots, and slightly higher at STD plots. While the rates of decline were similar between years, protease showed a continuous decline with depth (levelling at 30–40 cm at LTD), rather than the u-shaped pattern typically seen in Autumn 2014 (Table 2) and Spring 2015 (Table 3); urease declines were similar between years at UD and LTD, but at STD dropped rapidly to 10–20 cm and rose slowly from 30–50 cm; while arginine levels at depth remained very similar to those in Spring 2015, except for slightly lower topsoil levels and slightly higher subsoil levels (10–30 cm) at the STD plot.

There was often statistically significant correlation between the activities of protease and urease and the average $E_h$ of the preceding two weeks (Tables 5 – 7), especially in the Summer 2015 samples. The positive correlation between $E_h$ and enzyme activities points to higher microbial activity under less reducing conditions.



**Table 2.** Enzyme activity at the three mesotrophic study sites during Autumn 2014 (n=9; mean values ± standard error (SE); data subjected to Tukey HSD tests and Box-Cox transformation; different letters express statistically significant results, where lowercase letters (a) refer to mean comparisons of homogeneous groups vertically, and uppercase letters (A) refer to horizontally between surfaces.

| Treatment | Long-term Drainage (LTD) | Short-term Drainage (STD) | Undrained (UD) |
|---|---|---|---|
| Depth, cm | Protease | Protease | Protease |
| 0-10 | 219.4±11.4aA | 428.0±17.8aB | 1519.2±22.9aC |
| 10-20 | 128.1±14.4bA | 410.5±40.9abB | 590.7±12.1bC |
| 20-30 | 125.7±19.6bA | 347.8±17.5bB | 358.8±9.9cB |
| 30-40 | 217.4±23.9aA | 220.6±29.4cA | 324.7±14.0cB |
| 40-50 | 202.5±20.7aA | 198.5±26.4cA | 314.5±27.7cB |
| Depth, cm | Urease | Urease | Urease |
| 0-10 | 127.2±14.6aA | 193.0±17.2aB | 1119.6±10.1aC |
| 10-20 | 96.6±8.2aA | 74.5±11.3bA | 339.9±68.4bB |
| 20-30 | 37.9±3.1bA | 49.0±7.1bcA | 66.4±19.1cA |
| 30-40 | 27.2±4.0bcA | 39.0±3.5cA | 71.4±11.3cB |
| 40-50 | 19.7±6.0cA | 41.4±6.4cA | 35.1±8.1cA |
| Depth, cm | Arginine deaminase | Arginine deaminase | Arginine deaminase |
| 0-10 | 33.5±5.1aA | 46.6±8.3aA | 31.8±2.6aA |
| 10-20 | 30.5±2.9aA | 50.8±4.7aB | 27.4±1.7aA |
| 20-30 | 42.8±6.9aAB | 58.1±5.1aB | 31.8±1.1aC |
| 30-40 | 32.6±2.4aA | 59.9±1.7aB | 56.5±3.5bB |
| 40-50 | 50.2±5.7aA | 108.6±11.4bC | 72.5±5.9bB |



**Table 3.** Enzyme activity at the three mesotrophic study sites during Spring 2015 (n=9; mean values ± standard error (SE); data subjected to Tukey HSD tests and Box-Cox transformation; different letters express statistically significant results, where lowercase letters (a) refer to mean comparisons of homogeneous groups vertically, and uppercase letters (A) refer to horizontally between surfaces.

| Treatment | Long-term Drainage (LTD) | Short-term Drainage (STD) | Undrained (UD) |
|---|---|---|---|
| Depth | Protease | Protease | Protease |
| 0-10 | 282.8±7.2aA | 498.4±21.0aB | 1065.7±34.5aC |
| 10-20 | 164.6±10.2bA | 377.1±17.8bB | 700.6±24.2bC |
| 20-30 | 62.7±6.4cA | 281.0±20.6cB | 336.8±12.2cB |
| 30-40 | 113.9±7.9dA | 191.1±10.2dB | 298.4±15.9cdC |
| 40-50 | 213.6±8.4eA | 206.8±21.4dA | 247.1±16.2dA |
| Depth | Urease | Urease | Urease |
| 0-10 | 353.8±25.8aA | 446.8±14.9aB | 1366.6±25.9aC |
| 10-20 | 66.2±9.7bA | 38.6±7.4cA | 180.7±8.1bB |
| 20-30 | 27.9±7.6cdA | 53.5±6.0bB | 143.1±9.6cC |
| 30-40 | 39.4±2.9bcA | 17.6±3.7cB | 53.7±5.3dA |
| 40-50 | 17.4±3.1dA | 47.9±3.2cB | 53.5±3.7dB |
| Depth | Arginine deaminase | Arginine deaminase | Arginine deaminase |
| 0-10 | 11.0±0.6aA | 12.2±1.2abA | 7.4±0.3aB |
| 10-20 | 4.2±0.3cA | 6.2±0.2cB | 4.4±0.5cA |
| 20-30 | 3.2±0.6cA | 7.1±0.5cB | 8.4±0.4aB |
| 30-40 | 9.0±0.8abA | 10.0±0.4bAB | 11.1±0.4bB |
| 40-50 | 7.0±0.5bA | 13.1±0.4aC | 10.9±0.2bB |




**Table 4.** Enzyme activity at the three mesotrophic study sites during Summer 2015 (n=9; mean values ± standard error (SE); data subjected to Tukey HSD tests and Box-Cox transformation; different letters express statistically significant results, where lowercase letters (a) refer to mean comparisons of homogeneous groups vertically, and uppercase letters (A) refer to horizontally between surfaces.

| Treatment | Long-term Drainage (LTD) | Short-term Drainage (STD) | Undrained (UD) |
|---|---|---|---|
| Depth | Protease | Protease | Protease |
| 0-10 | 296.7±13.1aA | 700.1±20.1aB | 1605.6±34.9aC |
| 10-20 | 255.3±3.6bA | 362.3±14.9bB | 786.5±30.0bC |
| 20-30 | 115.1±5.2cA | 350.0±11.4bB | 308.3±14.6cB |
| 30-40 | 133.8±5.7cA | 271.1±11.3cB | 360.9±17.8cC |
| 40-50 | 133.9±5.0cA | 256.2±11.0cC | 214.95±10.3dB |
| Depth | Urease | Urease | Urease |
| 0-10 | 431.0±13.7aB | 162.5±7.7aA | 1124.3±13.8aC |
| 10-20 | 90.6±5.6bB | 40.9±3.5bA | 147.7±8.3bC |
| 20-30 | 71.5±4.9bB | 40.8±4.5bA | 53.2±2.8cA |
| 30-40 | 25.3±2.7cB | 48.8±2.5bA | 47.9±2.4cA |
| 40-50 | 6.9±1.2dA | 49.7±2.3bC | 35.3±2.9dB |
| Depth | Arginine deaminase | Arginine deaminase | Arginine deaminase |
| 0-10 | 12.8±0.8aA | 8.8±1.0aB | 7.5±0.4aB |
| 10-20 | 6.0±0.8bA | 9.4±0.6aB | 8.6±0.6abB |
| 20-30 | 7.6±0.3cA | 9.1±0.3aB | 8.4±0.3abAB |
| 30-40 | 7.3±0.3cB | 10.8±0.6aA | 10.3±0.1cA |
| 40-50 | 7.6±0.7cA | 13.9±0.4bB | 9.3±0.5bcA |

**Table 5.** Correlations of $E_h$ and enzyme activities within profiles, between depths. Year 2014 – Autumn, Mesotrophic fen. Treatment LTD and STD. Value for $E_h$ is the mean of two weeks preceding the sampling. Correlations are significant at p < 0.05, N=5. Statistically significant correlations are designated **.

| Variable-Treatment | | | | | | | |
|---|---|---|---|---|---|---|---|
| $E_h$-LTD | 1 | | | | | | |
| Protease-LTD | -0.5999 | 1.0000 | | | | | |
| Urease-LTD | 0.7478 | -0.0198 | 1.0000 | | | | |
| Arg. deam.-LTD | -0.4915 | 0.0133 | -0.6312 | 1.0000 | | | |
| $E_h$-STD | 0.4543 | 0.4167 | 0.8642 | -0.4329 | 1.0000 | | |
| Protease-STD | **0.9645**** | -0.4144 | **0.8956**** | -0.5579 | 0.6454 | 1.0000 | |
| Urease-STD | 0.5318 | 0.3229 | **0.8883**** | -0.4012 | **0.9933**** | 0.7087 | 1.0000 |
| Arg. deam.-STD | -0.7859 | 0.2471 | -0.6702 | 0.8563 | -0.4869 | -0.7738 | -0.4998 |




**Table 6.** Correlations of $E_h$ and enzyme activities within profiles, between depths. Year 2015 – Spring, Mesotrophic fen. Treatment LTD, STD and UD. $E_h$ is the mean of two weeks preceding the sampling. Correlations are significant at $p < 0.05$, N=5. Statistically significant correlations are designated **.

| Variable-Treatment | | | | | | | | | | | |
|---|---|---|---|---|---|---|---|---|---|---|---|
| $E_h$-LTD | 1 | | | | | | | | | | |
| Protease-LTD | 0.875 | 1 | | | | | | | | | |
| Urease-LTD | 0.9178** | 0.7456 | 1 | | | | | | | | |
| Arg. deam.-LTD | 0.5342 | 0.6857 | 0.6831 | 1 | | | | | | | |
| $E_h$-STD | 0.9482** | 0.7124 | 0.878 | 0.4201 | 1 | | | | | | |
| Protease-STD | 0.9102** | 0.5976 | 0.8679 | 0.2638 | 0.9587** | 1 | | | | | |
| Urease-STD | 0.8961** | 0.7527 | 0.9855** | 0.6694 | 0.8115 | 0.8296 | 1 | | | | |
| Arg. deam.-STD | 0.3318 | 0.686 | 0.3718 | 0.7731 | 0.0561 | -0.0465 | 0.4511 | 1 | | | |
| $E_h$-UD | 0.9794** | 0.7774 | 0.9449** | 0.5228 | 0.9832** | 0.9529** | 0.9021** | 0.2036 | 1 | | |
| Protease-UD | 0.9425** | 0.6774 | 0.9125** | 0.4261 | 0.9924** | 0.9773** | 0.8554 | 0.0523 | 0.9879** | 1 | |
| Urease-UD | 0.9098** | 0.7316 | 0.9955** | 0.6465 | 0.8552 | 0.8694 | 0.9951** | 0.3758 | 0.9308** | 0.8968** | 1 |
| Arg. deam.-UD | -0.5038 | -0.1413 | -0.3124 | 0.3562 | -0.685 | -0.734 | -0.2301 | 0.6173 | -0.5655 | -0.6622 | -0.3038 |





**Table 7.** Correlations of $E_h$ and enzyme activities within profiles, between depths. Year 2015 – Summer, Mesotrophic fen. Treatment LTD, STD and UD. Value for $E_h$ is the mean of two weeks preceding the sampling. Correlations are significant at $p < 0.05$, N=5. Statistically significant correlations are designated **.

| Variable-Treatment | $E_h$-LTD | Protease-LTD | Urease-LTD | Arg. deam.-LTD | $E_h$-STD | Protease-STD | Urease-STD | Arg. deam.-STD | $E_h$-UD | Protease-UD | Urease-UD | Arg. deam.-UD |
|---|---|---|---|---|---|---|---|---|---|---|---|---|
| $E_h$-LTD | 1 | | | | | | | | | | | |
| Protease-LTD | 0.9360** | 1 | | | | | | | | | | |
| Urease-LTD | 0.9499** | 0.8062 | 1 | | | | | | | | | |
| Arg. deam.-LTD | 0.8056 | 0.5478 | 0.9141** | 1 | | | | | | | | |
| $E_h$-STD | 0.9705** | 0.8264 | 0.9879** | 0.9233** | 1 | | | | | | | |
| Protease-STD | 0.9431** | 0.8079 | 0.9973** | 0.8936** | 0.9767** | 1 | | | | | | |
| Urease-STD | 0.9105** | 0.7128 | 0.9629** | 0.9750** | 0.9807** | 0.9424** | 1 | | | | | |
| Arg. deam.-STD | -0.4707 | -0.5141 | -0.5736 | -0.3016 | -0.4545 | -0.6145 | -0.3646 | 1 | | | | |
| $E_h$-UD | 0.8984** | 0.6922 | 0.9777** | 0.9777** | 0.9769** | 0.9670** | 0.9883** | -0.4377 | 1 | | | |
| Protease-UD | 0.9866** | 0.9292** | 0.9636** | 0.7979 | 0.9638** | 0.9604** | 0.9031** | -0.5939 | 0.8974** | 1 | | |
| Urease-UD | 0.9552** | 0.7976 | 0.9913** | 0.9393** | 0.9978** | 0.9795** | 0.9874** | -0.4727 | 0.9854** | 0.9559** | 1 | |
| Arg. deam.-UD | -0.747 | -0.6761 | -0.7848 | -0.6128 | -0.7375 | -0.8224 | -0.6433 | 0.5906 | -0.7322 | -0.7385 | -0.7233 | 1 |





### 3.4 Statistical modelling of redox potential

285 The results of modelling proved inconclusive. It was not possible to predict momentary $E_h$, or even the momentary change or direction of change in $E_h$ based on momentary WTL, time elapsed since WTL had risen above a given level or the direction of WTL change, even when combined with soil temperature and precipitation.

While it was possible to fit the non-linear model to some inundation periods, mostly on the ME-LTD plot at 25 cm depth, more often than not $E_h$ showed no consistent behaviour during inundation. Furthermore, the predictive power of the linear 290 models proved extremely poor when autocorrelation of $E_h$ was taken into account.

For further description of the models, please see Appendix A.

## 4 Discussion

### 4.1 WTL and redox

Though predicting $E_h$ using WTL proved impossible based on the models we applied, our results show a clear effect of WTL 295 regime on $E_h$ in terms of depth distribution. At the ME-LTD, ME-STD and OM-LTD plots, $E_h$ varied widely between 5 and 15 cm depth (and also at 25 cm depth in the case of the LTD plots), while at the ME-UD plot, variability was mostly limited to 5 cm depth (Fig. 3).

Generally speaking, differences between sensors were more pronounced when average $E_h$ was between the extremes, i.e. $O_2$-reducing and $CO_2$-reducing conditions, and when $E_h$ was in a state of change (Fig 3). This emphasises the highly localised 300 nature of redox conditions and the fact that the values at each Pt sensor only represent the conditions prevalent in a small volume of soil.

The overall $E_h$ values at different depths could be interpreted as indicating different TEAs being used at different depths in the peat matrix. In this case, $E_h$ values over 400mV, mostly present at 5 cm in all treatments, would be indicative of oxic conditions, values of $+100\ldots+400$ at 15cm depth would indicate $NO_3^-$ reduction and $E_h$ values at 25 cm depth and deeper 305 would be more indicative of $Fe^{3+}$, $SO_4^{2-}$ or $CO_2$ reduction (Table 1). Just such a connection between dynamic WTL and $Fe^{2+}$ and $SO_4^{2+}$ concentrations in porewater (indicating $E_h$ values below +50 and -100 mV, respectively) has previously been observed in a degraded fen in Central Europe, confirming that WTL does indeed have an effect on $E_h$ conditions, even if current measurements and analysis methods fail to record the connection.

The effect of WTL on $E_h$ was complex. One might expect a pattern where WTL rising over a certain level would, in all 310 cases, cause conditions to become more reducing as water blocks movement of air into the soil matrix. This type of relationship was previously observed by Mchergui et al. (2014) in laboratory studies on silt and sandy mineral soils under both permanent and intermittent flooding. In their study, the settling time for $E_h$ varied between 0 and 14 days, depending on soil type. On peatlands, de Mars and Wassen (1999) reported a clear relationship between groundwater level and $E_h$ values at 15 cm depth, especially on moderately rich fens. Note, however, that measurements were taken at many sites with only $2 - 4$ time points 315 measured per site; hence, variation in $E_h$ values at a given WTL was in the range of $200 - 400$ mV throughout the WTL range.



Our continuous measurements suggested that factors in addition to WTL were affecting the $E_h$, and that the relationship between WTL and $E_h$ was not always simple. For example, in the Summer of 2015 at the LTD plot, $E_h$ at 25 cm depth, where most changes in $E_h$ were observed, responded directly to changes in WTL, even though WTL at the time was below -30 cm. At the same time, there was no change in $E_h$ below 25cm depth, even when the WTL dropped below -50cm. At the UD plot, there were several occasions where a rising WTL caused a rise in $E_h$, e.g. after snow melt in Spring and Summer 2016 (Fig. 13). This could be an indication of incoming water bringing in new TEAs from the surrounding area. A similar trend of $E_h$ increasing with rising WTL was observed by Yu et al. (2008) in a swamp environment. In this case, the major difference to our site was that the WTL was, on average, much higher than the soil surface. It was assumed by the authors that the rise in $E_h$ was caused by $O_2$ transport to the rooting zone by plants. As the UD plot in our study was dominated by aerenchymous vegetation, such as sedges and wetland herbs (see section 2.1), this could also be the case in the current experiment. That the effect is limited to 5 cm depth is consistent with the maximal oxygen release depth observed for *Carex rostrata* (Mainiero and Kazda, 2005).

Over the same period, rain events caused a rapid drop in $E_h$ at 15 and 25cm depth, despite the WTL never rising above 40cm. This suggests that infiltration of rain water down to the WTL followng heavy rainfall causes enough blockage of gas transport to prevent reoxidisation of the TEAs used by the microbial population.

The difference in $E_h$ dynamics at the ME plots between winters appeared to be related to differences in WTL dynamics (Fig. 3). During the winter of 2014-2015, for example, the WTL fluctuated between -30 and -10 cm at the LTD plot, and between -20 and -5 at the STD and UD plots; whereas the WTL remained constant during the winter of 2015-2016, lying either close to the surface or above it at all sites. This may have been due to a brief period of frost early in the winter, when up to 20 cm of snow accumulated, followed by thawing over about a month, followed by freezing conditions once again. This could have formed a layer of ice on the surface of the peat, preventing flow of $O_2$ into the system. The fact that the deeper snow pack during the winter of 2014-2015 failed to cause reducing conditions in the top layers suggests that $O_2$ movement into peat is not necessarily prevented by snow. As redox conditions are intrinsically linked with water chemistry (e.g. Zak et al., 2004; Kjaergaard et al., 2012; Riedel et al., 2013; Kaila et al., 2016), this suggests that there may be differences in Spring melt runoff water quality and $CH_4$ emissions between years, depending on whether gas transport between the atmosphere and soil has been blocked by ice or not. Surface, as opposed to deep, peat through-flow, indicative of frozen top soil, has previously been associated with a higher $CH_4$ content (Dinsmore et al., 2011), as well as lower dissolved $O_2$ concentrations in Spring runoff water (Eskelinen et al., 2016).

To the best of our knowledge, multi-year studies of Spring melt-water quality and greenhouse gas emissions in relation to soil frost conditions have yet to be conducted. Nevertheless, the dynamics between soil frost and $E_h$ observed in this study (Fig 3) suggest that pore-water chemical quality under the snow pack and its relationship to $E_h$ measurements could be an interesting subject for future study.

The complexity of the WTL-$E_h$ relationship revealed by wavelet analysis most likely explains why the non-dynamic linear and non-linear models failed to predict changes in $E_h$ using momentary WTL or duration of inundation. Furthermore, there



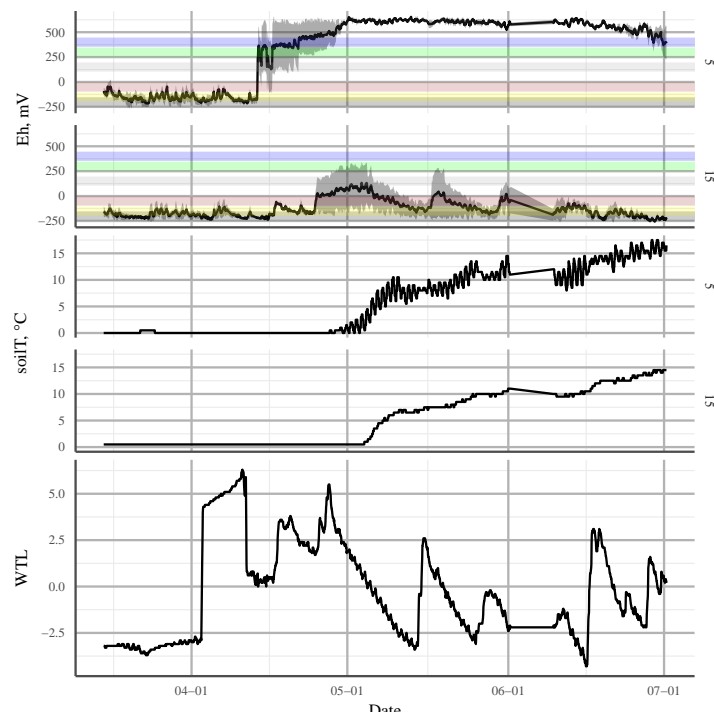

**Figure 13.** Redox potential ($E_h$) and soil temperature (soilT) at 5 and 15 cm depth, and water table level (WTL) at the undrained (UD) plot in Spring 2016. Solid black line in $E_h$: mean of 3 profiles, shaded area: S.E. Horizontal coloured bars represent approximate theoretical $E_h$ values of transition between different TEAs, from top to bottom: $O_2$, $NO_3^-$, $Mn^{4+}$, $Fe^{3+}$, $SO_4^{2-}$, $CO_2$

was no sign of $E_h$ reacting more rapidly to changes in WTL at the OM-LTD plot than the ME plot, whether using Eh0 isopotential-WTL wavelet analysis or a comparisons of daily $\delta E_h$ value distribution.

The difficulties faced in characterising $E_h$ behaviour based on WTL could, at least in part, be understood by defining exactly what WTL measured from a WTL well represents. It is easy to mistake the clear interface between air and the water surface in a well as representing a sudden switch from dry to wet conditions in the peat. In fact, the water level in the well represents

the depth at which water content in the peat reaches field capacity, ie. the degree of saturation where all pores small enough to allow capillary action to hold on to water against the force of gravity are filled. Depending on the peat's pore size distribution, there will still be varying amounts of non-saturated pores present in the peat at field capacity, and the proportion of saturated pores will decrease linearly above the water level (Zhang and Furman, 2021). Thus, not only can the actual soil water content at the WTL differ between sites and treatments, it may actually still increase below the WTL. As saturation of pores with water



is the main obstacle preventing $O_2$ from penetrating the soil profile, soil water content and pore size distribution at different depths could prove a better means of predicting $E_h$ behaviour in peatlands, and soils in general, than simply measuring the WTL. Our observation of no clear-cut correlation between $E_h$ and WTL could, in part, explain the often poor predictive power of WTL in momentary $CH_4$ dynamics (e.g. Koskinen et al., 2016; Korkiakoski et al., 2017) as $CH_4$ production and oxidation are intrinsically connected with $E_h$ conditions.

### 4.2 Enzyme activity

The variations in enzyme activity observed at the ME plots between Autumn 2014 and Summer 2015 were potentially the result of local differences in organic substrate quality and, more specifically, differences in the activity of hydrolytic enzymes at different points and depths at each site.

Generally speaking, in more oxidative environments, i.e. those with higher $E_h$, enzyme activity increased (Tables 5 − 7).
This was almost universally true for protease and urease activity, though not for arginine deaminase. This connection between hydrolytic enzymes and $E_h$ indicates that degradation of complex organic substances is inhibited under reducing conditions, which would promote carbon accumulation in the soil. Interestingly, this general increase in enzyme activity under more oxic conditions was also confirmed by Alef and Kleiner (1987a) for arginine deaminases, the author's also stating that aerobic and anaerobic ammonification of arginine (including their ratio) was most likely strongly influenced by the redox potential.
Consequently, the non-correlation between arginine deaminase and redox potential observed in our study may have been due to the inhibition of arginine deaminase, perhaps by an increased amount of readily available substrate for microorganisms and a high amount of $NH_4$ in the soil (Alef and Kleiner, 1987b; Lin and Brookes, 1999). The potential for ammonification by microorganisms could also have been limited by a higher C/N ratio in the soil environment (Fujii et al., 2019). While we are unable to deduce the exact cause of this unexpected arginine deaminase inhibition from our results, redox potential could prove
to be an indicator for this condition.

There are two other possible explanations for the differences observed, both related to water regime. Phenol oxidases are one of the few enzyme groups capable of breaking down phenolic compounds, and any reduction in their activity, i.e. their ability to degrade phenolic compounds, may explain the reduced activity of hydrolytic enzymes (Bonnett et al., 2006). Phenol oxidases are a group of extracellular enzymes that catalyse the oxidation of lignin and phenolic compounds in soil organic
matter (Sanchez-Julia and Turner, 2021). The influence of such oxidising enzymes on the activity of hydrolytic enzymes has been described by Freeman et al. (2001) using the 'enzyme latch' hypothesis, which states that "under conditions of full saturation with water, there is a lack of $O_2$ in the environment, which suppresses the activity of oxidising enzymes". Furthermore, the low levels of $O_2$ naturally found in peat bogs and fens leads to the accumulation of polyphenolic substances, which bind to hydrolytic enzymes and thereby inhibit their functioning (Fenner and Freeman, 2011). However, Romanowicz
et al. (2015) found that fluctuations in the WTL can cause changes in the accumulation of polyphenolic substances (i.e. phenol oxidase activity) in peat soils, as well as redox potential, with both redox potential and phenol oxidase activity decreasing with increasing WTL. Up until now, however, studies on the impact of water level on extracellular enzymes have produced



conflicting results (e.g. Xiang and Freeman, 2009; Sun et al., 2010; Fenner et al., 2011), and further research will be needed to clarify the processes involved.

A second possibility is the 'iron gate' paradigm described by Wang et al. (2017), whereby the effect of phenolic substances on the oxidation, and subsequent mineralisation, of soil organic carbon (SOC) following a drop in water levels (water balance) is attributed to Fe transformation, which is redox induced (Li et al., 2012). In this case, Fe(II) is oxidised to Fe(III) through abiotic or biotic reactions (Hall and Silver, 2013). As Fe(II) positively affects the oxidation of phenols (Sinsabaugh, 2010; Hall and Silver, 2013), Fe oxidation may serve to protect against C loss as the WTL drops (Wang et al., 2017). Under conditions

of reduced moisture, oxidation of Fe(II) will reduce the oxidation of phenols and thereby affect the mineralisation rate of SOC (Wang et al., 2017), thus indirectly affecting the activity of hydrolytic enzymes. The Fe oxidation process also lowers the pH, which will also affect phenol oxidase activity as its optimal pH is close to neutral (Sinsabaugh, 2010; Toberman et al., 2010). Furthermore, the products of Fe(II) oxidation can immobilise P and dissolved organic C, thereby reducing C and P availability, and thus enzyme synthesis (Lalonde et al., 2012; Riedel et al., 2013). Extracellular enzymes can also bind to amorphous ferric

hydroxide ($Fe(OH)_3$) formed during Fe(II) oxidation, further limiting the catalytic activity of enzymes in peat soil (Allison, 2006; Wen et al., 2019).

## 5   Conclusions

To the best of our knowledge, this is the first study to undertake long-term continuous monitoring of redox conditions at multiple depths in a boreal peatland, while also monitoring soil WTLs and temperatures. Furthermore, the side-by-side existence of

different drainage histories on plots with similar ecohydrological development before drainage makes our dataset unique.

   Our results showed that enzyme activity changed at different depths and in different seasons, most likely due to differences in substrate quality at the different sites, particularly in relation to water saturation. Such effects were particularly pronounced where water saturation was directly affected by the drainage regime.

   Our first hypothesis, that reduced levels of alternate TEAs in the OM-LTD plot would cause Eh conditions to become

reducing more quickly after a rise in the WTL was not confirmed by our data, which indicated that alternative TEAs were utilised in drained peatlands with more decomposed peat.

   Our second hypothesis, that $E_h$ and Fe reduction isopotential fluctuations would closely follow WTL fluctuations, was only partially supported by the data. The connection between WTL and $E_h$ was intermittently in syn- and counter-phase, suggesting that different hydrological phenomena may affect redox conditions in opposite ways, even where the effect on WTL is constant

in all cases.

   Our third hypothesis, that the OM drained plot would display more rapid $E_h$ changes when the WTL fluctuated, was not supported by the data. It is possible that, in this case, the greater OM-associated electron accepting capacity of decomposed *Sphagnum* peat, as opposed to sedge peat, compensated for the lack of mineral elements.

   Our final hypothesis, that $E_h$ measurements at the OM drained plot would display a bi-modal distribution, and those in the

ME drained plot a multi-modal distribution due to the greater availability of mineral TEAs, was confirmed. This suggests that,



if the electron accepting capacity of organic matter is slowing down the transition between oxic and $CO_2$-reducing conditions, its $E_h$ may not be picked up by the sensor's $Pt$ electrodes.

Finally, the relationship between WTL fluctuation and redox conditions proved too complex to model with non-dynamic linear or non-linear modelling, suggesting that characterisation of redox fluctuations between treatments will require a more 430 complex model.

We suggest that future field $E_h$ measurement campaigns should be paired with pore water chemistry and greenhouse gas monitoring, and that the role of OM and changes in input water quality in the yearly redox cycle should be clarified.

*Data availability.* The original ($E_h$, WTL, soilT, enzymatic activity, pH) data is available in Zenodo: 10.5281/zenodo.12544806. Other data and scripts are available from corresponding author upon reasonable request.

**Appendix A: Modelling results**

**A1   Methods**

For the nonlinear models (Eq. A1), it was assumed that different sites and depths in the peat profile would have characteristically low $E_h$ values toward which the system would asymptotically move under inundated conditions, i.e. when movement of $O_2$ into the peat matrix was limited or prevented. It was also assumed that $E_h$ values would rise rapidly once inundation ended as 440 oxygen penetration into the soil profile oxidised the reduced TEAs. Consequently, an exponential asymptotic model was fitted separately to the mean measurements from each peat depth on all treatments using the time inundated (i.e. the time elapsed since WTL had risen above each Pt sensor in the peat profile) as the explanatory variable. Each period of inundation was fit separately, with a lower limit of 240 hours for inundation length to exclude short-term WTL fluctuations that may not be long enough for the $\delta E_h$ to show signs of saturation. In this way, characteristic asymptote and rate values were obtained for each 445 depth and treatment.

$$E_h = A_0 + (A_{ref} - A_0) * (1 - e^{-t_{in}/\tau}) \tag{A1}$$

where $E_h$ is the redox potential measured, $A_0$ is $E_h$ at the beginning of the inundation period, $A_{ref}$ is the asymptotic term of $E_h$, $t_{in}$ is the time inundated (hours) and $\tau$ is the rate term. Model fitting was undertaken in the R statistical environment (R Core Team, 2023) using the Levenburg-Marquardt-algorithm and the minpack.lm library (Elzhov et al., 2022).

A multiple linear model was also fit to each treatment and sensor depth using available ancillary measurements as parameters (Table A1). Autocorrelation between consecutive $E_h$ measurements was then estimated using the ar function in R, with a lag value of 1. The autocorrelation was then entered into the linear model fitting procedure.



**Table A1.** Parameters used in the linear models

| Name | explanation | unit |
|------|-------------|------|
| $D_o$ | depth of WTL over the sensor | cm |
| $Pr_d$ | running sum of precipitation, 24h | mm |
| $Pr_w$ | running sum of precipitation, 7 days | mm |
| $T_{soil}$ | temperature of soil at sensor depth | °C |
| $T_{air}$ | temperature of air at 4.2m height (at SMEAR II station) | °C |
| $I_{in}$ | time since WTL rose above sensor depth | h |

## A2  Results

It was only possible to fit the asymptotic model to 36% of inundation periods with a continuous duration of 240 hours or more. Grouped by treatment and depth, fitting success rate varied between 0 and 56% of inundation periods for those groups where the number of viable periods available was > 1.

More often than not, the $E_h$ value did not react to inundation in a monotonic fashion, which was the main assumption behind use of an asymptotic function to model the connection between time inundated and $E_h$. In practice, $E_h$ values often fluctuated around the Fe reduction level or higher, reaching levels indicative of $O_2$ reduction, even during months-long inundation periods.

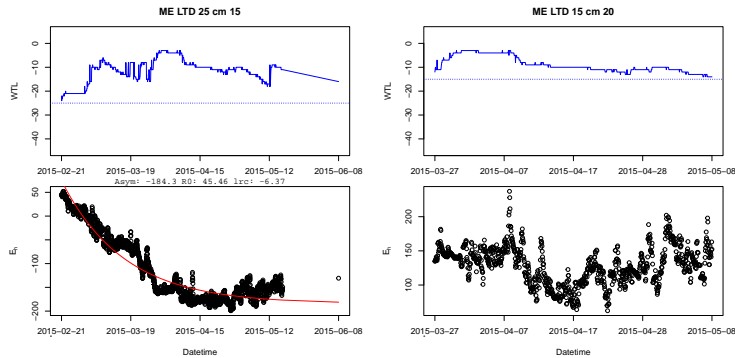

**Figure A1.** Example of a successful (left) and failed (right) fit of the asymptotic model to $E_h$ during periods of inundation. Top panels, solid blue line: water table level, dashed blue line: position of $E_h$ sensor in the depth profile. Left Figures: Asym is the value of the asymptote, R0 the $E_h$ value when inundation began and lrc the natural logarithm of the rate constant (mV s$^{-1}$)

Even though the linear models were fitted for each depth at each treatment separately, for the most part they completely failed to reproduce the temporal dynamics of $E_h$ (Figs. A2, A3). Only on the UD plot, at 35 cm depth, was the average $E_h$ value predicted well and even then any shorter term temporal dynamics were completely missing (Fig. A4).



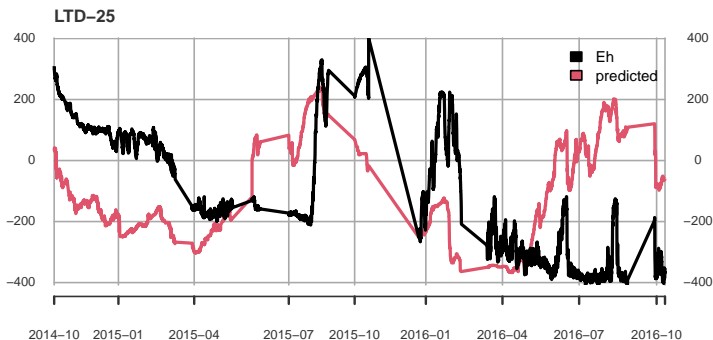

**Figure A2.** Measured (black) and predicted (red) redox potential at ME-LTD plot, 25 cm depth.

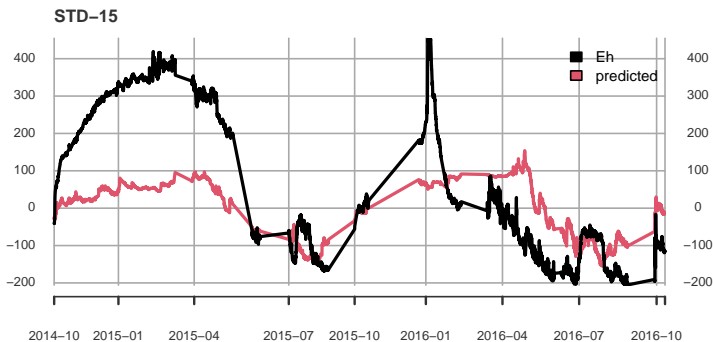

**Figure A3.** Measured (black) and predicted (red) redox potential at ME-STD plot, 15 cm depth.

## A3 Conclusions

In the data presented here, momentary WTL, soil and air temperature and historical precipitation were not sufficient to reliably

explain the temporal changes in $E_h$. Possibly some sort of dynamic model could work better in predicting how $E_h$ is affected

by environmental variables over time.



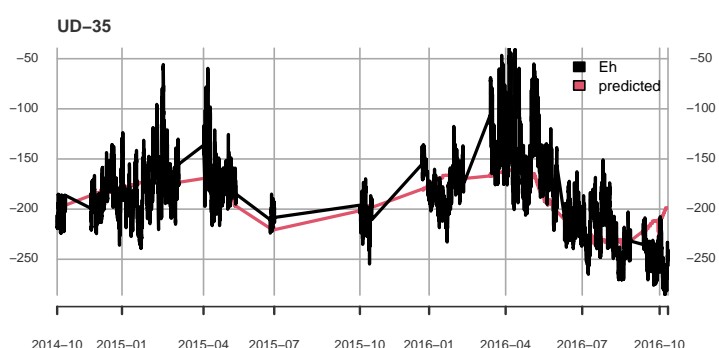

**Figure A4.** Measured (black) and predicted (red) redox potential at ME-UD plot, 35 cm depth.



*Author contributions.* MK processed and analysed the data and did all calculations and model fitting unless otherwise noted, wrote the first draft and revised the manuscript with input from all co-authors; JA did the wavelet analysis of hourly data; VV and LH performed the field work, analysis and interpretation of the enzyme activity assessment and wrote the subsections concerning enzymatic activity; KR performed English proofreading; MV designed, built and installed the field redox potential measurement system; RL conceived the original research

idea, planned the field experiment and supervised the project.

*Competing interests.* The authors declare no competing interests

*Acknowledgements.* The authors wish to thank Petra Strakova for help with peat sampling and chemical analysis. This work was supported by the Finnish Research Council (Grant no's 289116 and 339489) and the Maj and Tor Nessling Foundation.



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
