# Peer review of "Covariation of redox potential profiles and water table level in peatland sites representing different drainage regimes: implications for ecological modelling"

_EGUsphere, 2024_

## Referee Comment (RC1)

**Referee comments Koskingen et al.** *Covariation of redox potential profiles and water table level in peatland sites representing different drainage regimes: implications for ecological modelling*

**General comments**

The manuscript features the undervalued but fundamental redox potential that serves as indicator of peat mineralization processes. Besides, the redox potential is coupled to measurements of enzyme activity. The authors raise interesting hypotheses and present valuable results (although figures and tables might still be distilled better). The results are well-integrated in literature in the discussion, and conclusions are communicated clearly. Nevertheless, I believe that certain improvements should be made before publication. In particular, I have concerns regarding the methodology used to determine the Fe-reduction isopotential (and the associated hypothesis), as well as the discussion regarding bi-modality in Eh (and the answers provided to this hypothesis). Please find more specific comments below.

**Major comments**

*Methodology*

$E_h$ and reduction thresholds are impacted by pH, therefore it is important to mention the pH when referring to Eh values or reduction ranges (in text and tables). In fact, the authors normalized the Eh measurements for pH 7 by applying a correction slope of 59.2 mV pH$^{-1}$ (Nernst equation). This correction may become problematic for the Eh threshold of Fe reduction, as the correction slope differs for Fe (177.6 mV pH$^{-1}$) as protons are involved in half-reactions. This is depicted in Pourbaix diagrams for iron. At a common peat soil pH of 5 and a certain Eh, the iron in the soil might be reduced according to iron Pourbaix diagrams, but after normalization with a slope of 59.2 mV pH$^{-1}$ to pH 7, the normalized Eh instead might indicate that the iron in the soil is oxidized. Therefore, I would either suggest to normalize the Eh to a common pH value (closer to the average of soil pH measurements), or to use a different Eh correction slope when assessing the Fe-reducing isopotential. As a result, answers to the associated hypothesis might change.

*Results*

As a reader, it is difficult to find the subplot that is referred to in the text. The amount of figures and subfigures is quite high. I would recommend to reduce the amount of figures and subplots showing wavelet coherences. For example, Fig. 4-6 show very similar

results, one of these figures is enough for the reader to understand patterns in wavelet temperature coherence (the results that are similar could be moved to the supplements). WTL wavelet coherences could be represented within a separate figure. Furthermore, it would be very helpful to include direct references within the figures in such a way that the reader directly understands which variables/probes/study sites are represented by a subplot (or subplot row or column). Additionally, I would suggest to combine correlation tables and/or move some correlation tables to the appendix.

*Discussion*

The authors raise the hypothesis that the redox potential shows bi-modal behaviour at the ombrotrophic plot, and that more nutrient rich conditions result in a multi-modal distribution. Probability plots (Fig. 12) confirm this hypothesis. However, I believe that the results are insufficiently placed into context of groundwater level fluctuations in the discussion. In fact, the groundwater level is much more stable at the OM plot compared to the ME plot, which would also result in less Eh variability (also see Boonman et al. 2024, https://doi.org/10.1016/j.geoderma.2023.116728 ) and a higher likeliness of bi-modal behaviour ($CO_2$ reduction below the groundwater level, $O_2$ reduction above). Based on this nuance, I think that the hypothesis about bi-modality cannot be confirmed.

*Conclusion & hypotheses*

The first and third hypothesis in the conclusion seem to be similar. Furthermore, the arrangement of hypotheses in the introduction deviates from the arrangement in the conclusions. The research actually features many hypothesis which is sometimes confusing. Perhaps some of the hypotheses could be combined or the hypotheses could be restructured. Also, it would be nice if implications of study outcomes could be added to the conclusion section.

**Minor comments**

Line 89: Because generally more TEA's are present in minerotrophic peatlands.

Table 1: The Eh values and ranges presented in the table lack referencing (and the pH value at which these Eh values and ranges were determined).

Line 123: "potentially bringing in electron acceptors such as Fe to the mesotrophic (ME) plots." Have measurements been done that confirm this?

Line 307: Please add a reference.

Line 314: "Note, however…". Does this sentence refer to Mars and Wassen (1999)?

Line 315: For relations between Eh and groundwater level, also see Boonman et al. (2024, https://doi.org/10.1016/j.geoderma.2023.116728 )

Line 360: See also Estop-Aragones et al. (2012, http://dx.doi.org/10.1029/2011JG001888 ) for discussion on saturated pores, redox potential and oxygen presence.

---

## Author Comment (AC1)

**Author's response to reviewers' comments on manuscript EGUSPHERE-2024-2050**

**General**

We thank the reviewers for their comments. Below we have outlined our responses to their question, suggestions and requests. You will find that for the most part we agree with the comments and will improve our manuscript accordingly; where we disagree with them, we have written out our reasons for it, which hopefully justifies our point of view.

**Reviewer 1**

**Major comments**

**Methodology**

- Comment  Eh and reduction thresholds are impacted by pH, therefore it is important to mention the pH when referring to Eh values or reduction ranges (in text and tables). In fact, the authors normalized the Eh measurements for pH 7 by applying a correction slope of 59.2 mV pH-1 (Nernst equation). This correction may become problematic for the Eh threshold of Fe reduction, as the correction slope differs for Fe (177.6 mV pH-1) as protons are involved in half-reactions. This is depicted in Pourbaix diagrams for iron. At a common peat soil pH of 5 and a certain Eh, the iron in the soil might be reduced according to iron Pourbaix diagrams, but after normalization with a slope of 59.2 mV pH-1 to pH 7, the normalized Eh instead might indicate that the iron in the soil is oxidized. Therefore, I would either suggest to normalize the Eh to a common pH value (closer to the average of soil pH measurements), or to use a different Eh correction slope when assessing the Fe-reducing isopotential. As a result, answers to the associated hypothesis might change.

- Reply  As evidenced by Fig. 12 [densityplot], we had a significant cluster of Eh observations around 0 mV on the ME LTD plot, where iron could be expected to be the major inorganic non-oxygen electron acceptor. We did not see peaks in observation density at other possible iron reduction Eh values.

  0 mV as indicator of iron reduction is also supported by textbook literature, such as Stumm and Morgan (1996) referenced by eg. Borch et al.

(2010). We also have empirical lab measurements done as part of a master's thesis (Marttunen, 2024), where iron addition to anaerobic peat stabilised the pH 7-normalised Eh value measured with Pt electrodes and Ag/AgCl reference electrodes to aroud 0 mV.

Due to these reasons we respectfully decline the reviewer's suggestion.

**Results**

- **Comment** As a reader, it is difficult to find the subplot that is referred to in the text. The amount of figures and subfigures is quite high. I would recommend to reduce the amount of figures and subplots showing wavelet coherences. For example, Fig. 4-6 show very similar results, one of these figures is enough for the reader to understand patterns in wavelet temperature coherence (the results that are similar could be moved to the supplements). WTL wavelet coherences could be represented within a separate figure. Furthermore, it would be very helpful to include direct references within the figures in such a way that the reader directly understands which variables/probes/study sites are represented by a subplot (or subplot row or column). Additionally, I would suggest to combine correlation tables and/or move some correlation tables to the appendix.

- **Reply** We thank the reviewer for these suggestions for improving the readability and understandability of our manuscript. We've come to the conclusion that it's enough to show the WTL wavelets and temperature wavelets at 15 cm depth from the three ME plots in one figure, which will be a new Figure 4. This will reduce the number of wavelet figures by 2. The Eh wavelet figures we'll leave as is, as they highlight the different responses between the parallel probes on each plot. Furthermore, we will increase the font size in and label the subplots according to what plot, parameter and depth (if applicable) they represent.

  Regarding the enzyme correlation tables, we agree with the reviewer and will combine the tables into one table which contains the frequency of significant correlation between the variables, depending on how many times they were measured on each plot (i.e. $0 \ldots 3/3$ or $0 \ldots 2/2$). We will also make the enzyme activity tables into graphical form, where activities at each depth will be represented by mean $\pm$ S.D. in a depth profile plot.

  The original, per-measurement campaign enzyme activity tables, enzyme correlation tables and all Eh wavelet figures will be put into Supplements.

**Discussion**

- **Comment** The authors raise the hypothesis that the redox potential shows bi-modal behaviour atthe ombrotrophic plot, and that more nutrient rich conditions result in a multi-modal distribution. Probability plots (Fig. 12) confirm this hypothesis. However, I believe that the results are insufficiently placed into context of groundwater level fluctuations in the discussion. In fact, the groundwater level is much more stable at the OM plot compared to the ME plot, which would also result in less Eh variability (also see Boonman et al.2024, `https://doi.org/10.1016/j.geoderma.2023.116728` ) and a higher likeliness of bi-modal behaviour

(CO2 reduction below the groundwater level, O2 reduction above). Based on this nuance, I think that the hypothesis about bi-modality cannot be confirmed.

- Reply   Here again we must respectfully disagree with the reviewer. In Figure 12, we show the density of Eh values from the probes placed at 25cm depth on both the ME-LTD and the OM-LTD plots. Iron reduction and oxidation are important during periods of transition from oxic to anoxic conditions and vice versa, respectively, which should occur when the WTL raises to above or drops to below the depth of interest, which is here 25cm. On the ME-LTD plot, although the WTL varies over a wider range of depths than on the OM-LTD plot (SD of WTL is 20cm on the ME-LTD plot, compared to 7 cm on the OM-LTD plot), there are as a matter of fact fewer transitions of WTL from below 25cm to above than on the OM-LTD plot (14 versus 26 transitions, respectively), which would suggest that the OM-LTD plot should be more prone to multimodal Eh than the ME-LTD plot, which is contrary to our observations. Thus our rejection of the null hypothesis, that there would be no multimodality vs. bimodality on the ME and OM plots, respectively, stands.

  We will add this reasoning to the discussion to support our conclusions.

**Conclusions & hypotheses**

- Comment   The first and third hypothesis in the conclusion seem to be similar. Furthermore, the arrangement of hypotheses in the introduction deviates from the arrangement in the conclusions. The research actually features many hypothesis which is sometimes confusing. Perhaps some of the hypotheses could be combined or the hypotheses could be restructured. Also, it would be nice if implications of study outcomes could be added to the conclusion section.

- Reply   We apologize for the confusing arrangement of and discrepancies between the hypotheses in the Introduction and in the Conclusions. We will alter the Introduction to properly attribute ordinal numbers to the hypotheses, so that the paragraph on lines 97-100 becomes the first hypothesis, spllit the paragraph on lines 101-105 to two parts first stating the tested relationships and the other outlining the second hypothesis. Further we will number the current second, third and fourth hypotheses as the third, fourth and fifth hypotheses, respectively (on lines 106-109, 110-112 and 113-115 in the original manuscript, respectively). The we will address these hypotheses in the Conclusions in the same order.

  The first hypothesis is supported by the data as shown in Fig. 3 and described on lines 227-232, where the most dynamic Eh is observed closest to the surface at the ME-UD plot, furthest down at the ME-LTD plot, with the ME-STD plot between these two, reflecting the dominant WTL at each plot.

  The second hypothesis is partially supported by the data, as shown in Fig. 11, where there are periods of significant coherence in the waveforms between the WTL and Eh 0 isopotential in all plots except ME-STD.

  The third hypothesis is not supported by the data.

The fourth hypothesis is supported by the data, as shown in Fig. 12. where a bimodal distribution in the Eh is seen at the OM-LTD plot at 25cm depth, contrasting with the three-peaked distribution in Eh at the ME-LTD plot.

The fifth hypothesis is not supported by the data, namely the samples from the UD plot with generally lower Eh values than the drained plots displayed generally higher enzyme activities than the drained plots; however, within plot a higher Eh predicted higher enzyme activity.

**Minor comments**

1. Line 89: Because generally more TEA's are present in minerotrophic peatlands.

   **R:** Good point, we will add this clarification to the manuscript.

2. Table 1: The Eh values and ranges presented in the table lack referencing (and the pH value at which these Eh values and ranges were determined).

   **R:** We will add references to the table. Eh values are by default presented at pH7, but we will add the information to the caption.

3. Line 123: "potentially bringing in electron acceptors such as Fe to the mesotrophic (ME) plots." Have measurements been done that confirm this?

   **R:** We will add a reference to a symposium proceedings where chemical characteristics of water from the esker is presented (Sallantaus and Kaipainen, 1996).

4. Line 307: Please add a reference.

   **R:** We will add the reference omitted by mistake by us (Estop-Aragonés et al., 2013)

5. Line 314: "Note, however...". Does this sentence refer to Mars and Wassen (1999)?

   **R:** It does; we will clarify this in the text.

6. Line 315: For relations between Eh and groundwater level, also see Boonman et al. (2024, `https://doi.org/10.1016/j.geoderma.2023.116728` )

7. Line 360: See also Estop-Aragones et al. (2012, `http://dx.doi.org/10.1029/2011JG001888` ) for discussion on saturated pores, redox potential and oxygen presence.

   **R** to 6 and 7: Thank you for pointing out these articles; we will add them to our Discussion.

**Reviewer 2**

**Minor comments**

1. Lines 110-115: It was interesting to see a focus on hydrolytic enzymes, given the focus on Eh. Why were no oxidative enzymes chosen? It seems

this would be more relevant to 'enzymic latch' mentioned in discussion? I think it would be appropriate to justify that here.

**Response:** We chose to focus on enzymes that break down nitrogen compounds and thus affect nitrogen availability. We were aware of possible limitations in the activities of hydrolytic enzymes from the papers by Freeman et al. (2001) or Bonnett et al. (2006) and assumed that some limitations in activities would occur. We did not measure oxidative enzymes because we did not want to repeat results that other authors had. With hindsight, it might have been appropriate to measure oxidative enzyme activity, but we did not consider it useful at the time. The referee is right that it would have been relevant to have these results and then discuss them in the discussion section.

Regarding the targeting of hydrolytic enzymes and Eh, we did not find relevant studies on this topic in the literature. Therefore, we have omitted this topic in the discussion section, also in view of the lack of data on oxidative enzyme activities.

2. Lines ˜157-162; Line 185: There should be some more information about the "redox probes". Were these calibrated in any way (run against a standard, like Zobell's or Lights Solution). How were they conditioned? Were the pH electrodes calibrated, and how?

**R:** The probes are described in Vorenhout et al. (2011). We will add a reference to the article earlier in the text and add a short description of the probes.

3. Table 1, Figure 13 bars: I appreciate the theoretical thermodynamic cut-off ranges for redox pairs for context, but seeing as the authors did not measure redox couples in this study, perhaps this could be removed, or put in appendix? Just a suggestion.

**R:** Here we respectfully disagree with the reviewer. The different redox pairs and the ability to estimate the dominant one at a given time are a central motivation to study the useability and predictability of redox potential in soil ecosystems. Thus the typical redox pairs in soil systems and their redox potentials are essential background information for the readers.

4. Line 170: Again, interesting focus on hydrolytic, given the focus on oxidation reduction potential.

**R:** We believe this point was addressed in out reply to comment #1.

5. Line 182: Why was 200mV added to the probe reading? The authors later state that Eh-pH was normalized using the Nernst equation; what is this "+200mV" doing?

**R:** The +200mV correction is done to compensate for the differences in electron activity between the Ag/AgCl refernce electrode which are commonly used in field studies, and the hydrogen electrode against which Eh values are reported. We will add this clarification to the text.

6. Figure 3: It is surprising that there is not really a vertical temperature gradient.

**R:** There is a temperature gradient visible in Figure 3, but it is mostly present during summer. During winter when the temperature of water relatively quickly reaches close to 0 celsius but due to the heat released in phase change doesn't easily drop below that, temperature gradient becomes less steep.

7. Wavelet analysis figures: I think the captions could make these more accessible. What does the Y axis represent? I apologize for not having expertise in this area, but I suspect a large amount of the readership of this paper won't either.

   **R:** We will clarify these very good points raised in the comment. Y axis represents the frequency of a wavelet, in hours in these figures.

8. Enzyme activity tables: Please provide units and provide context in the text (results and discussion). As written, we do not know if these values are "reasonable"; are these potential activities in the pocket of prior research?

   **R:** We apologize for omitting the units from the tables and will add them to the text. The values are within an order of magnitude as soil enzyme activities reported in e.g. (Lloyd and Sheaffe, 1973; Singh and Kumar, 2008; Wojciech and Styła, 2011).

   It must be noted that the activities of these enzymes have rarely been estimated in peatlands. We will however add a short comparison of our results and earlier work to the text.

   Units:

   - Protease - µg L-tyrosine $h^{-1}$ $g^{-1}$ soil d.w.
   - Urease - µg NH4-N $h^{-1}$ $g^{-1}$ soil d.w.
   - Arginine deaminase - µg NH4-N $h^{-1}$ $g^{-1}$ soil d.w.

9. Correlation Matrix Tables: It would be helpful if the x axis could also contain labels, so the reader doesn't have to count columns to compare r values.

   **R:** We will thoroughly rework the correlation tables. We will reduce the number of tables to one, which will present the frequency of significant correlation between variables (i.e. 0-2/2 or 0-3/3, depending on how many measurements were made). We will also add labels to the X axis of the table.

10. Discussion: I am very interested in the analysis performed that is buried in the Appendix (A1-A3).

    **R:** As stated in the main text, the results of the modelling excercise were disappointing. Most of the time the predicted redox potential represented a different dominant redox pair than the measurements. It is possible that the whole concept of predicting the redox potential is misguided, or that experiments under controlled conditions should be done in order to tease apart the effects of growing season stage, temporal environmental conditions and inputs from the watershed.

We will add a short meditation on the subject into the discussion, exploring the problems we faced and possible solutions, such as models predicting the probability of a given redox pair dominating the redox conditions versus peat properties and environmental factors, as well as dynamic approach utilising reservoirs of electron acceptors.

11. Lines ~325-330 : It has also been shown that meteoric inputs from rainfall can have an oxidizing effect on pore water chemistry. Please see the paper by Mitchell and Branfireun (2005) in the journal, "Ecosystems", 8:731-747.

    **R:** This is a good point to add to the discussion, thank you! We will compare our observations with Mitchell and Branfireun. It must be noted that the highest effect on Eh in that study was found on the upland-peatland interface, whereas in our study the plots were well within the peatland (see Fig. 1 in the manuscript.)

12. Line ~345: Moreover, temperature plays a critical role in the solubility of Oxygen in water (see your appendix models). It is common to see a gradual reduction in DO with the progression of the growing season in peatlands, as soils warm and respiration increases. I think a brief nod to this would be good, and this would be a good place to engage your linear and nonlinear modelling exercises.

    **R:** The solubility of Oxygen in water changes between temperatures of 5 and 20 celsius by 3/12 mg L-1, or by 25%. Whether this could have a significant effect on how fast respiration by microbes consumes the Oxygen and causes anoxia and especially getting parameter estimates for the size of the effect would require a controlled condition experiment rather than a field measurement campaign. We will add a brief discussion on this to the manuscript and update our recommendations for future experiments to include reductionist approaches for restraining parameter estimates of redox state change.

13. Figures A1-A3: I would like to see perhaps XY predicted vs. observed plots; if you squint here, it actually looks like these more traditional approaches are doing fairly well at predicting Eh.

    **R:** We will add such plots along the time series comparisons.

**References**

Bonnett, S.A.F., Ostle, N., Freeman, C., 2006. Seasonal variations in decomposition processes in a valley-bottom riparian peatland. Science of The Total Environment 370, 561–573. doi:10.1016/j.scitotenv.2006.08.032.

Borch, T., Kretzschmar, R., Kappler, A., Cappellen, P.V., Ginder-Vogel, M., Voegelin, A., Campbell, K., 2010. Biogeochemical Redox Processes and their Impact on Contaminant Dynamics. Environmental Science & Technology 44, 15–23. doi:10.1021/es9026248.

Estop-Aragonés, C., Knorr, K.H., Blodau, C., 2013. Belowground in situ redox dynamics and methanogenesis recovery in a degraded fen during dry-wet cycles and flooding. Biogeosciences 10, 421–436. doi:10.5194/bg-10-421-2013.

Freeman, C., Ostle, N., Kang, H., 2001. An enzymic 'latch' on a global carbon store. Nature 409, 149. doi:10.1038/35051650.

Lloyd, A.B., Sheaffe, M.J., 1973. Urease activity in soils. Plant and Soil 39, 71–80. doi:10.1007/BF00018046.

Marttunen, S., 2024. Impacts of Controlled Redox Conditions on Greenhouse Gas Dynamics from Peat. Master's thesis. University of Helsinki, Faculty of Biological and Environmental Sciences. Helsinki, Finland.

Sallantaus, T., Kaipainen, H., 1996. Water-carried element balances of peatlands. Northern peatlands in global climatic change, edited by: Laiho, R., Laine, J., and Vasander, H., Publications of the Academy of Finland, Edita, Helsinki , 197–203.

Singh, D.K., Kumar, S., 2008. Nitrate reductase, arginine deaminase, urease and dehydrogenase activities in natural soil (ridges with forest) and in cotton soil after acetamiprid treatments. Chemosphere 71, 412–418. doi:10.1016/j.chemosphere.2007.11.005.

Stumm, W., Morgan, P., 1996. Water Chemistry. volume 3.

Vorenhout, M., van der Geest, H.G., Hunting, E.R., 2011. An improved datalogger and novel probes for continuous redox measurements in wetlands. International Journal of Environmental Analytical Chemistry 91, 801–810. doi:10.1080/03067319.2010.535123.

Wojciech, S.L., Styła, K., 2011. Changes of Urease Activity in Peat Profile of Peatland By Nierybno Lake in "Bory Tucholskie" National Park , 4.